# Duality and mock modularity

**Atish Dabholkar[1], Pavel Putrov[1] and Edward Witten[2]**

**1** International Centre for Theoretical Physics, Strada Costiera 11, Trieste 34151 Italy
**2** School of Natural Sciences, Institute for Advanced Study,
Einstein Drive, Princeton, NJ 08540, USA

## Abstract

We derive a holomorphic anomaly equation for the Vafa-Witten partition function for twisted four-dimensional $\mathcal{N} = 4$ super Yang-Mills theory on $\mathbb{CP}^2$ for the gauge group $SO(3)$ from the path integral of the effective theory on the Coulomb branch. The holomorphic kernel of this equation, which receives contributions only from the instantons, is not modular but 'mock modular'. The partition function has correct modular properties expected from $S$-duality only after including the anomalous nonholomorphic boundary contributions from anti-instantons. Using M-theory duality, we relate this phenomenon to the holomorphic anomaly of the elliptic genus of a two-dimensional noncompact sigma model and compute it independently in two dimensions. The anomaly both in four and in two dimensions can be traced to a topological term in the effective action of six-dimensional $(2, 0)$ theory on the tensor branch. We consider generalizations to other manifolds and other gauge groups to show that mock modularity is generic and essential for exhibiting duality when the relevant field space is noncompact.



# 1   Introduction

The hypothesis of $S$-duality asserts that $\mathcal{N} = 4$ super Yang-Mills theory is invariant under the action of a large duality group ($SL(2,\mathbb{Z})$ or a close relative, depending on the four-dimensional gauge group $G$) acting on $\tau \equiv \tau_1 + i\tau_2 = \theta/2\pi + 4\pi i/g^2$; here $g$ and $\theta$ are the gauge coupling and theta angle; $\tau_1$ and $\tau_2$ denote the real and imaginary parts of $\tau$ throughout this paper. But $S$-duality is hard to test, because computations for strong coupling are difficult. One way to circumvent this difficulty is to consider a topologically twisted version of the theory in which localization can be used to perform computations for strong coupling.

In this paper, we will consider one particular twisting, originally studied in the present context in [1]. With this twisting, a formal argument shows that the partition function on a compact four-manifold $X$ is holomorphic in $\tau$ or equivalently in $q = \exp(2\pi i\tau)$. Furthermore, if a certain curvature condition (eqn. (2.58) in [1]) is satisfied, the evaluation of the path integral can formally be argued to localize on the contribution of ordinary Yang-Mills instantons. (Without this curvature condition, one localizes on the solutions of a more complicated system of equations.) The contribution to the path integral from the component of field space with instanton number[1] $n$ is then $a_n q^n$, where $a_n$ is the Euler characteristic of the instanton number $n$ moduli space $\mathcal{M}_n$. Thus the partition function after summing over bundles of all

---

[1]Here $n$ is an integer for a simply-connected gauge group such as $G = SU(2)$, but may have a fractional part if $G$ is not simply-connected. The fractional part is determined by a two-dimensional cohomology class (for example, by the second Stieffel-Whitney class $w_2$ if $G = SO(3)$), and in the partition function $\sum_n a_n q^n$, it is natural to sum over all bundles keeping this class fixed. The values of $n$ in the sum are then congruent to each other mod $\mathbb{Z}$. A restriction on $w_2$ (and its analog for other groups) is assumed in eqn. (1) and other similar formulas in this paper.

values of the instanton number is expected to be

$$Z = \sum_n a_n q^n. \tag{1}$$

The relevant curvature condition is highly restrictive, but there are a number of four-manifolds that satisfy this condition and for which computations of the $a_n$ were available in the mathematical literature [2,3]. In particular, two important examples are a K3 surface and $\mathbb{CP}^2$. For K3, the expectations were borne out; the function $\sum_n a_n q^n$ is a holomorphic modular function[2]. What happens for $\mathbb{CP}^2$ is more complicated. There is a natural modular function $Z(\tau, \bar{\tau})$ whose holomorphic part is $\sum_n a_n q^n$, but this function is not holomorphic. It has a "holomorphic anomaly" which for $SO(3)$ bundles with $w_2 = 0$ reads[3]

$$\frac{\partial Z}{\partial \bar{\tau}} = \frac{3}{16\pi i \tau_2^{3/2} \eta(\tau)^3} \sum_{n \in \mathbb{Z}} \bar{q}^{n^2}. \tag{2}$$

(There is a second such formula, which we consider later, for bundles with nonzero $w_2$.) A microscopic explanation of the failure of holomorphy was not provided in [1]. However, it was noted that the right hand side of eqn. (2) – and also its analog with $w_2 \neq 0$ – looks like it could come from a sum over abelian anti-instantons on the Coulomb branch. On the Coulomb branch, the gauge group is broken from $SO(3)$ (or $SU(2)$) to $U(1)$, and $\bar{q}^{n^2}$ can be interpreted as the exponential of the classical action for a $U(1)$ anti-instanton of flux $n$. On this interpretation, the origin of the factor $\frac{3}{16\pi i} \tau_2^{-3/2}$ is not immediately apparent. It is anyway not clear why we should be summing over anti-instantons in a theory that formally can be argued to localize on instantons.

Subsequent developments have made it clear that the holomorphic anomaly must indeed come from the Coulomb branch and more specifically from a surface term at infinity on the Coulomb branch. One development involves Donaldson theory of four-manifolds or more precisely its interpretation in terms of $\mathcal{N} = 2$ super Yang-Mills theory. A formal argument shows that certain correlation functions in a twisted version of the $\mathcal{N} = 2$ theory depend only on the smooth structure of a four-manifold $X$ and not on its Riemannian metric $g$. These correlators are expected to coincide with the Donaldson invariants. From a mathematical point of view [4], the Donaldson invariants are true invariants for $b_2^+ > 1$ but for $b_2^+ = 1$, they instead have a chamber structure: they are generically invariant under a small change in the metric $g$, but they jump when one crosses certain "walls" in the space of metrics. This phenomenon is analogous to wall-crossing for BPS states in various supersymmetric models. The wall-crossing phenomenon was studied in [5] from a gauge theory point of view and was found to originate from a surface term at infinity on the Coulomb branch.[4] In other words, the formal proof that certain correlation functions are independent of $g$ involves integration by parts in field space. Upon "localization," the proof requires integration by parts on the Coulomb branch of the theory, and there is a possibility of a surface term at infinity. Such a surface term arises for $b_2^+ = 1$ and accounts for wall crossing.

Going back to $\mathcal{N} = 4$, the formal proof of holomorphy of the twisted theory again involves integration by parts, so it is reasonable to ask if again there may be a surface term at infinity

---

[2]Here and below by a "modular function" we mean a function which is invariant under the action of a congruence subgroup of $SL(2,\mathbb{Z})$, up to a possible multiplier system. For the reasons explained later, in this paper the partition function is assumed to be normalized so that it has modular weight zero.

[3]Up to a factor of 2, this formula also holds for gauge group $SU(2)$.

[4]For $b_2^+ > 1$, there are enough fermion zero-modes on the Coulomb branch to prevent wall crossing behavior, so the Donaldson invariants are true topological invariants. There has been very little study of the case $b_2^+ = 0$, partly because nonzero Donaldson invariants for gauge group $SU(2)$ or $SO(3)$ (the groups most studied) can only arise if $b_1 + b_2^+$ is odd, so that if $b_2^+ = 0$, $b_1$ must be nonzero. But most of the interest in four-manifold theory is on simply-connected four-manifolds, which necessarily have $b_1 = 0$. However, the case $b_2^+ = 0$ certainly merits more study.

on the Coulomb branch that accounts for the holomorphic anomaly. Indeed, it was pointed out in [1] that $\mathbb{CP}^2$ has $b_2^+ = 1$ (while K3 has $b_2^+ > 1$) and it was suggested that a holomorphic anomaly would arise on any four-manifold with $b_2^+ = 1$. The goal of the present paper is to demonstrate this, by performing the appropriate analog of the computation in [5].

Before saying more about this, we pause to explain a dual version of the problem. A two-dimensional supersymmetric field theory of a rather general type has a natural invariant, the elliptic genus. It is defined by a path integral on a torus $T^2$. Fields in the sigma-model are taken to be periodic functions on the torus up to possible twists by symmetries; the twists are chosen so that the supercurrent associated to one of the supersymmetries is invariant, but otherwise one allows arbitrary twists. For a sigma-model with a compact target space (or for any supersymmetric field theory with a discrete spectrum), the elliptic genus is a holomorphic function of the modular parameter $\tau$. However, for a sigma-model with a noncompact target space, the elliptic genus can have a holomorphic anomaly [6–15]. The elliptic genus defined by a path integral on the torus is then still modular invariant, but it is no longer holomorphic. In this situation, the elliptic genus becomes, in modern language, a mock modular function rather than an ordinary holomorphic modular function.

In a sigma-model with target $W$, supersymmetric localization reduces the computation of the elliptic genus to an integral over the space of constant maps from $T^2$ to $W$. This space of constant maps plays the role of the Coulomb branch in the gauge theory. The space of such constant maps is a copy of $W$. The proof of holomorphy involves an integration by parts on $W$, and the anomaly in holomorphy comes from a surface term at infinity. For sigma-models, this has been studied in a variety of ways in the literature. The derivation in [14], with a direct calculation of the holomorphic anomaly in terms of the behavior at infinity in $W$, will be particularly useful as background to our computation.

The two arenas for a holomorphic anomaly that we have mentioned – gauge theory in four dimensions and supersymmetric field theory in two dimensions – can be dual, for the following reason. $S$-duality in four dimensions is believed to be intimately connected with the existence and properties of a certain superconformal field theory in six dimensions, the $(2, 0)$ model. In particular, the $(2, 0)$ theory on Euclidean six manifold $M = T^2 \times X$, in the limit that the area of the $T^2$ is very small, keeping fixed its complex structure, is expected to reduce to $\mathcal{N} = 4$ super Yang-Mills theory on $X$, with the $\tau$ parameter of the gauge theory simply equal to the $\tau$ parameter that determines the complex structure of $T^2$. The twisting of the $\mathcal{N} = 4$ theory on $X$ that is under discussion here can be "lifted" to a twisting of the $(2, 0)$ model on $T^2 \times X$. Formally, the partition function of this twisted version of the theory should not depend on the metric of $X$ and should depend holomorphically on $\tau$. But we will be exploring a possible anomaly in this holomorphy.

We will study the $(2, 0)$ theory on $T^2 \times X$ in either of two limits. If the $T^2$ is very small compared to $X$, then as already stated, we reduce to gauge theory on $X$. We will call this the gauge theory region. In the opposite limit that $X$ is very small compared to $T^2$, we reduce to a supersymmetric (possibly superconformal) field theory on $T^2$. We will call this the sigma-model region, since the supersymmetric model in question can be described as a sigma-model in the asymptotic region of field space that is important for the holomorphic anomaly (something as simple as this is not expected in the interior). Because the area of the $T^2$ does not matter in the topologically twisted theory, we must get the same holomorphic anomaly whether we compute in the sigma-model region or the gauge theory region.

In the gauge theory region, we will exhibit the holomorphic anomaly by a computation somewhat analogous to that in [5], and in the sigma-model region, we will exhibit the same anomaly by a calculation somewhat along lines of [14]. Actually, a computation using only the lowest order terms in the effective action on the Coulomb branch of the gauge theory or the target space of the sigma-model will not show the holomorphic anomaly. In that approx-

imation, the holomorphic anomaly vanishes. It is necessary to include a certain correction in the effective action. In understanding wall crossing in $\mathcal{N} = 2$ super Yang-Mills theory, the 1-loop quantum correction to the classical metric on the Coulomb branch plays an important role. For $\mathcal{N} = 4$, there is no such quantum correction to the metric of the Coulomb branch, but at the 1-loop level, a half-BPS correction to the effective action on the Coulomb branch is generated [16]. By exploiting holomorphy or a relation to anomalies, it can be shown that the coefficient of this interaction is 1-loop exact. It turns out that this interaction has the right properties to generate the expected holomorphic anomaly in the gauge theory approach.

The six-dimensional $(2, 0)$ model on its Coulomb branch likewise has a half-BPS coupling, first described in [17, 18], that reduces after $T^2$ compactification to the half-BPS interaction of the $\mathcal{N} = 4$ super Yang-Mills theory that was already mentioned. This interaction as well has a precisely known coefficient. We will show that this 6d coupling, after twisted compactification on $X$, has just the right properties to generate the expected holomorphic anomaly in the sigma-model approach.

The computations of the holomorphic anomaly both in the gauge theory region and the sigma model region yield the same result on the right hand side of (2), but the various factors have different origins in the two regions.

- The factor of 3 is related to the first Chern class of the canonical line bundle of $\mathbb{CP}^2$ in the gauge theory region and to the quantum of $H$-flux in the sigma-model region.
- The factor of $\tau_2^{-3/2}$ comes from the integral over the constant mode of the auxiliary field in the gauge theory region and from the integral over three non-compact bosonic zero-modes in the sigma model region.
- The factor of $\eta(\tau)^{-3} = \eta(\tau)^{-\chi(\mathbb{CP}^2)}$ is the contribution of point-like instantons in the gauge theory region and of the left-moving oscillators in the sigma model region.
- Finally, the anti-holomorphic theta-function $\sum_{n \in \mathbb{Z}} \bar{q}^{n^2}$ is a contribution of abelian anti-instantons in the gauge theory region and of right-moving momenta of a compact chiral boson in the sigma model region.

In one important respect, our sigma model calculation in two dimensions is more complete than our corresponding gauge theory calculation. In the sigma model, we will have to do a path integral on a two-torus. Such a path integral can be interpreted as a Hilbert space trace, and this determines its absolute normalization. By contrast, in gauge theory we will be doing a Coulomb branch calculation on a general four-manifold $X$. Such a path integral does not have a natural normalization; it can be affected, for example, by topological terms proportional to the Euler characteristic and the signature of $X$. To determine the absolute normalization of the Coulomb branch path integral, we would have to start in the ultraviolet with conventions that lead to a holomorphic expansion of the precise form (1), and then deduce the resulting normalizations on the Coulomb branch. We will not attempt to do that.

Now we mention some previous and current work on related problems. Mock modularity arising from Coulomb branch integrals in gauge theories with $\mathcal{N} = 2$ supersymmetry has been systematically explored in [19], extending previous calculations that had been done by more special methods [5, 20–24]. Moreover, in forthcoming work, Manschot and Moore have analyzed the Coulomb branch integral and the associated mock modularity in the $\mathcal{N} = 2^*$ theory, which of course is closely related to $\mathcal{N} = 4$ super Yang-Mills, which we study in the present paper. Their calculation might lead to a way to resolve the normalization issue mentioned in the last paragraph.

We now comment on the relation between the holomorphic anomaly and mock modularity. The naive holomorphic partition function of the twisted $SO(3)$ super Yang-Mills theory on $\mathbb{CP}^2$ is the holomorphic kernel of the anomaly equation (2) which receives contributions only from the instantons. It is holomorphic but not modular. The presence of the holomorphic

anomaly implies that the physical partition function necessarily contains a nonholomorphic piece given by an Eichler integral of the anomaly [25] which receives contributions from the anti-instantons. In modern terminology [26–28], the holomorphic piece is a (vector valued) 'mixed mock modular form' whereas the anomaly is governed by its 'shadow'. The physical partition function satisfying the anomaly equation is the 'modular completion' and has good modular properties, as expected from duality.

These considerations extend naturally to other Kähler 4-manifolds with $b_2^+ = 1$, $b_1 = 0$ and to other groups. In general, when the configuration space of the twisted theory is noncompact, the partition function is modular but not holomorphic, and satisfies a holomorphic anomaly equation similar to (2). This incompatibility between holomorphy and modularity is the essence of mock modularity. The physical requirement of duality invariance of the path integral thus leads naturally to the mathematical formalism of mock modularity whenever the relevant configuration space is noncompact.

The structure of the paper is as follows. In Section 2 we review relevant facts about topologically twists of $\mathcal{N} = 4$ SYM theory, their M-theory realizations and some generalities about holomorphic anomaly. In Section 3 we derive the holomorphic anomaly equation for the $SO(3)$ super Yang-Mills theory on $\mathbb{CP}^2$ by a computation in the gauge theory region. In Section 4 we rederive the anomaly in corresponding sigma model region. The nonholomorphic contributions in both regions can be seen to originate from a topological term on the six-dimensional world volume theory of the Euclidean M5-brane described in Section 4.1. In Section 5 we present generalizations to other manifolds and other gauge groups.

## 2 Twisting and Topological Field Theory

The contents of this section are as follows. In Section 2.1, we review the general notion of a topologically twisted theory. In Section 2.2 we review all three possible twists of $\mathcal{N} = 4$ 4d super Yang-Mills theory. In Section 2.3 and Section 2.4 we comment on the geometric interpretation of the twists, including their realization in M-theory. In Section 2.5 we give a general discussion on the origin of the holomorphic anomaly in topologically twisted theories.

### 2.1 Generalities

Here we briefly recall how $\mathcal{N} = 4$ super Yang-Mills theory can be "topologically twisted" to make what formally is a topological field theory. We say "formally" because, as we will discuss, the proof of topological invariance always relies on integration by parts in field space, which can generate a surface term under some circumstances. For more details on some of the following, see for example [1].

We first consider the theory on $\mathbb{R}^4$. The rotation group in four dimensions, extended to encompass spin, is[5] $\text{Spin}(4) = SU(2)_\ell \times SU(2)_r$. The $R$-symmetry group of $\mathcal{N} = 4$ super Yang-Mills theory is $SU(4)_R$. The global supersymmetries transform under $SU(2)_\ell \times SU(2)_r \times SU(4)_R$ as $(\mathbf{2}, \mathbf{1}, \mathbf{4}) \oplus (\mathbf{1}, \mathbf{2}, \overline{\mathbf{4}})$, where representations are labeled in a familiar way.

For "twisting" in the sense that we will consider, one picks a homomorphism $\rho : \text{Spin}(4) \to SU(4)_R$. Then one defines a new group $\text{Spin}(4)'$, isomorphic to $\text{Spin}(4)$, that consists of elements of $\text{Spin}(4) \times SU(4)_R$ of the form $g \times \rho(g)$, $g \in \text{Spin}(4)$.

One picks $\rho$ so that the representation $(\mathbf{2}, \mathbf{1}, \mathbf{4}) \oplus (\mathbf{1}, \mathbf{2}, \overline{\mathbf{4}})$ of $\text{Spin}(4) \times SU(4)_R$ contains at least one $\text{Spin}(4)'$ singlet. Let us denote as $Q$ a supercharge of $\mathcal{N} = 4$ super Yang-Mills theory that is such a singlet. It will always obey $Q^2 = 0$. The reason is that, more generally,

---

[5]In our conventions, self-dual 2-forms such as $F^+$ transform in the representation $(\mathbf{3}, \mathbf{1})$ of $\text{Spin}(4) = SU(2)_\ell \times SU(2)_r$. Instanton configurations satisfy $F^+ = 0$ and hence are anti-self-dual.

if $Q$ is any linear combination of the global supercharges of $\mathcal{N} = 4$ super Yang-Mills theory, its square will be a linear combination of the translation generators,[6] and these (as they commute with $SU(4)_R$) transform the same way under Spin(4)′ as under Spin(4). In particular, no nonzero translation generator is Spin(4)′-invariant, so, if $Q$ is Spin(4)′-invariant, $Q^2$ must vanish. If there are multiple Spin(4)′-invariant supercharges $Q$ and $Q'$, this argument shows that $Q^2 = (Q')^2 = \{Q, Q'\} = 0$.

The basic idea of making a twisted topological field theory is now to view $Q$ as a BRST-like operator: we only consider operators and states that are $Q$-invariant, and we consider an operator $\mathcal{O}$ to be trivial if it is a $Q$-commutator, $\mathcal{O} = \{Q, \mathcal{O}'\}$ for some $\mathcal{O}'$, and a state $\Psi$ to be trivial if it is $Q$-exact, $\Psi = Q\Lambda$ for some $\Lambda$. Because $Q$ generates a symmetry of the path integral and $Q^2 = 0$, adding $Q$-exact terms to the operators or states will not affect the expectation values of $Q$-exact operators in $Q$-exact states. Specializing to $Q$-closed operators and states will lead to a topological field theory because in all cases (i.e. for every choice of $\rho$), the stress tensor, which measures the response of the theory to an infinitesimal change in a background metric, is $Q$-exact,

$$T_{\mu\nu} = \{Q, \Lambda_{\mu\nu}\}, \tag{3}$$

where $\Lambda_{\mu\nu}$ is a linear combination of components of the supercurrent of the theory. Equation (3) is a special case of the usual commutation relation of the supercharges and supercurrents, written in a way that is natural in the twisted theory.

So far we have considered the theory on $\mathbb{R}^4$. It turns out that it is possible to formulate the twisted theory on a rather general four-manifold $X$, preserving the $Q$ symmetry, as long as one imposes on $X$ a mild condition that depends on $\rho$ and is detailed below. In fact, the generalization to a curved four-manifold $X$ can be made in a way that preserves all of the Spin(4)′ invariant supercharges, and the fact that any linear combination of them squares to zero.

With the goal of formulating the theory on a general manifold, we view Spin(4)′ as the rotation symmetry group, so in general fields do not have the same spin they have under the original rotation group Spin(4). We use the Spin(4)′ quantum numbers in coupling the fields of the $\mathcal{N} = 4$ theory to a background curved metric on any manifold $X$. This defines the coupling to a curved background modulo some nonminimal terms (explicit coupling to the Riemann tensor of $X$) which in some cases are needed to preserve the Spin(4)′ invariant supersymmetries.

The coupling to the background curved metric can be made in such as way that eqn. (3) still holds, that is, the stress tensor remains $Q$-exact. Formally, this means we get a topological field theory: as long as we consider only $Q$-invariant operators and their matrix elements between $Q$-invariant states, the expectation value or matrix element of $\{Q, \Lambda_{\mu\nu}\}$ will vanish because of $Q$-invariance of the path integral, and hence the response of the theory to a change in the background metric of $X$ will vanish. However, as noted in the introduction, this step needs to be treated with care. The claim that $\langle\{Q, \mathcal{V}\}\rangle = 0$ for any operator $\mathcal{V}$ is ultimately based on integration by parts in field space. An anomaly might come from a surface term at infinity. For example, in Donaldson theory of four-manifolds, viewed as twisted $\mathcal{N} = 2$ super

---

[6]We do not need to consider central charges in the supersymmetry algebra for the following reason. For our application, we will view $Q$ and $Q^2$ as automorphisms of the algebra of local operators. Central charges commute with local operators, so we can ignore them. Central charges can always be defined by surface integrals at spatial infinity, so they likewise would not appear if we quantize the theory on a compact spatial three-manifold and try to define a space of physical states (it is natural to do this in topological field theory, but we will not actually do so in the present paper). Central charges can appear if we quantize the theory on a noncompact three-manifold with boundary conditions at infinity that correspond to spontaneous gauge symmetry breaking (this is natural physically, but is usually less interesting in the context of topological field theory and is rather distant from our interests in the present paper).

Yang-Mills theory, the relevant integral reduces to an integral on the Coulomb branch, and one does find an anomaly – a surface term at infinity on the Coulomb branch – that spoils topological invariance in the case $b_2^+(X) = 1$ [5]. As explained in the introduction, in the present paper we will find that a somewhat similar anomaly spoils holomorphy in twisted $\mathcal{N} = 4$ super Yang-Mills, again if $b_2^+(X) = 1$.

## 2.2 The Three Twisted Theories in Detail

Now let us describe in more detail the twisted theories of interest. In practice, there are three possible choices of the homomorphism $\rho : \mathrm{Spin}(4) \to SU(4)_R$, given that we want to have at least one $\mathrm{Spin}(4)'$-invariant supercharge $Q$:

(A) The choice of $\rho$ of primary interest in this paper is such that the $\mathbf{4}$ of $SU(4)_R$ transforms under $\mathrm{Spin}(4)' = SU(2)'_\ell \times SU(2)'_r$ as $(\mathbf{2}, \mathbf{1}) \oplus (\mathbf{2}, \mathbf{1})$. Thus $\mathrm{Spin}(4)'$ commutes with a residual subgroup $SU(2)_R \subset SU(4)_R$ which permutes the two $(\mathbf{2}, \mathbf{1})$'s, and the $\mathbf{4}$ of $SU(4)_R$ transforms under $SU(2)'_\ell \times SU(2)'_r \times SU(2)_R$ as $(\mathbf{2}, \mathbf{1}, \mathbf{2})$. The 16 global supersymmetries of the $\mathcal{N} = 4$ theory transform under $\mathrm{Spin}(4)' \times SU(2)_R = SU(2)'_\ell \times SU(2)'_r \times SU(2)_R$ as $(\mathbf{1}, \mathbf{1}, \mathbf{2}) \oplus (\mathbf{3}, \mathbf{1}, \mathbf{2}) \oplus (\mathbf{2}, \mathbf{2}, \mathbf{2})$. In particular, the $(\mathbf{1}, \mathbf{1}, \mathbf{2})$ is a pair of $\mathrm{Spin}(4)'$-invariant singlet supercharges. Because they transform as $\mathbf{2}$ of a residual global $R$-symmetry $SU(2)_R$, it does not matter which linear combination $Q$ we use in defining a topological field theory; all choices are equivalent up to the action of $SU(2)_R$.

(B) The choice of $\rho$ useful in applications to the geometric Langlands program [29] is such that the $\mathbf{4}$ of $SU(4)_R$ transforms under $\mathrm{Spin}(4)'$ as $(\mathbf{2}, \mathbf{1}) \oplus (\mathbf{1}, \mathbf{2})$. Thus $\mathrm{Spin}(4)'$ commutes with a residual $U(1)_R$ subgroup of $SU(4)_R$, which we normalize so that the $(\mathbf{2}, \mathbf{1})$ and $(\mathbf{1}, \mathbf{2})$ respectively have charges 1 and $-1$. The supercharges of the theory transform under $\mathrm{Spin}(4)' \times U(1)_R$ as $2(\mathbf{1}, \mathbf{1})_1 \oplus (\mathbf{3}, \mathbf{1})_1 \oplus (\mathbf{1}, \mathbf{3})_1 \oplus 2(\mathbf{2}, \mathbf{2})_{-1}$, where the subscript is the $U(1)_R$ charge, and a 2 in front means that a representation appears twice. In particular, there are two $\mathrm{Spin}(4)'$ singlets. They transform the same way (both with charge 1) under the unbroken global symmetry $U(1)_R$. We can choose $Q$ to be a linear combination $\alpha Q_1 + \beta Q_2$ of the two $\mathrm{Spin}(4)'$ singlets $Q_1$ and $Q_2$. The resulting family of topological field theories depends in a nontrivial fashion on the parameter $t = \alpha/\beta$, which plays an important role in the application to geometric Langlands.

(C) The last case is that the $\mathbf{4}$ of $SU(4)_R$ transforms under $\mathrm{Spin}(4)'$ as $(\mathbf{2}, \mathbf{1}) \oplus 2(\mathbf{1}, \mathbf{1})$. Thus $\mathrm{Spin}(4)'$ commutes with a residual $U(2)_R$ subgroup of $SU(4)_R$, under which the two copies of $(\mathbf{1}, \mathbf{1})$ transform as a doublet. The global supersymmetries transform under $SU(2)'_\ell \times SU(2)'_r \times U(2)_R$ as $(\mathbf{1}, \mathbf{1}, \mathbf{1})_{-1} \oplus (\mathbf{3}, \mathbf{1}, \mathbf{1})_{-1} \oplus (\mathbf{2}, \mathbf{2}, \mathbf{1})_1 \oplus (\mathbf{2}, \mathbf{1}, \mathbf{2})_1 \oplus (\mathbf{1}, \mathbf{2}, \mathbf{2})_{-1}$, where the subscript is the charge under the center of $U(2)_R$. In particular, there is up to scaling a unique $\mathrm{Spin}(4)'$ singlet global supercharge $Q$ that we can use to make a topological field theory.

In each of the three cases, it is straightforward to determine how the fields of $\mathcal{N} = 4$ super Yang-Mills theory transform under $\mathrm{Spin}(4)'$. In particular, let us look at the adjoint-valued scalar fields $\phi$ of the theory. Before twisting, they transform in the $\mathbf{6}$ of $SU(4)_R$. By examining how they transform in the twisted theory, we will find what condition must be placed on $X$ so that the twisted topological field theory can be defined on $X$. (No additional condition comes from the gauge field $A$, as it is $SU(4)_R$-singlet so it is not affected by twisting; and no additional condition comes from the fermions, essentially because they are related to the bosons by the action of $Q$ and so transform the same way under $\mathrm{Spin}(4)'$.)

(A$'$) With our first choice of $\rho$, the six scalars transform under $SU(2)'_\ell \times SU(2)'_r \times SU(2)_R$ as $(\mathbf{3}, \mathbf{1}, \mathbf{1}) \oplus (\mathbf{1}, \mathbf{1}, \mathbf{3})$. In particular, this representation is not invariant under exchange of $SU(2)'_\ell$ and $SU(2)'_r$, so the theory with this twist does not have a parity or reflection symmetry, and $X$ must be oriented. However, the twisted scalars have integer spin, as do the fermions after twisting, so there is no need for $X$ to have a spin structure. That is why we can take $X = \mathbb{CP}^2$, the case in which a holomorphic anomaly was found in [1].

(B$'$) With the second choice of $\rho$, the six scalars transform under $SU(2)'_\ell \times SU(2)'_r \times U(1)_R$ as $(\mathbf{2}, \mathbf{2})_0 \oplus (\mathbf{1}, \mathbf{1})_2 \oplus (\mathbf{1}, \mathbf{1})_{-2}$. This is a reflection symmetric representation, so $X$ need not be oriented. (If $X$ is unorientable, the parameter $t$ discussed earlier is no longer arbitrary, since orientation-reversal acts nontrivially on this parameter.) The twisted scalars have integer spin, so $X$ again does not require a spin structure and we can study this theory on $\mathbb{CP}^2$.

(C$'$) With the third choice of $\rho$, the six scalars transform under $SU(2)'_\ell \times SU(2)'_r \times U(2)_R$ as $(\mathbf{2}, \mathbf{1}, \mathbf{2})_1 \oplus (\mathbf{1}, \mathbf{1}, \mathbf{1})_2 \oplus (\mathbf{1}, \mathbf{1}, \mathbf{1})_{-2}$. This representation is not reflection symmetric, so $X$ must be oriented. In addition, some of the scalars have half-integer spin, so $X$ must carry a spin structure.

## 2.3 Geometrical Realization

In what follows, it will be useful to be familiar with a geometrical realization [30] of the three twisted theories. First let us consider a realization by D3-branes in Type IIB superstring theory. We can realize $\mathcal{N} = 4$ super Yang-Mills, with gauge group $U(N)$, by wrapping $N$ D3-branes on $X$. Of course, Type IIB superstring theory is naturally defined on a 10 dimensional spacetime $Y$, so $X$ will have a rank 6 normal bundle $\mathcal{W}$ in $Y$. The scalars in the super Yang-Mills multiplet describe normal oscillations of the D-branes, so they are valued in $\mathcal{W}$ tensored with the adjoint representation of the gauge group $G$. Since we have determined how the scalars transform under the symmetries, we can read off what must be the normal bundle to $X$ in $Y$:

(A$''$) With our first choice of $\rho$, three scalars transform under Spin$(4)'$ as $(\mathbf{3}, \mathbf{1})$. This is the appropriate representation for a selfdual second rank tensor or two-form. So one summand in $\mathcal{W}$ is the bundle $\Omega_2^+(X)$ of selfdual two-forms on $X$. The other scalars are Spin$(4)'$ singlets in the vector representation of $SU(2)_R$. The upshot of this is that we can take $Y$ to be $\Omega_+^2(X) \times \mathbb{R}^3$, where here by $\Omega^+_2(X)$ we mean the total space of the rank three vector bundle $\Omega_2^+(X) \to X$, and $\mathbb{R}^3$ is a copy of three-dimensional Euclidean space, with $SU(2)_R$ as its group of rotations. $X$ is embedded in $\Omega_+^2(X) \times \mathbb{R}^3$ as the zero-section of $\Omega_2^+(X)$, times a point in $\mathbb{R}^3$, which we can choose to be the origin.

For some favorable choices of $X$, such as $\mathbb{CP}^2$ or $S^4$, $\Omega_2^+(X)$ admits a complete metric of $G_2$ holonomy, such that the zero-section is a "coassociative" (or supersymmetric) submanifold. This puts the supersymmetry of the twisted model in a standard framework. For more generic $X$, there presumably is no nice complete metric of $G_2$ holonomy on $\Omega_2^+(X)$, but we can think of $\Omega_2^+(X)$ as carrying a $G_2$ structure near the zero-section. That is sufficient for purposes of gauge theory.

(B$''$) With the second choice of $\rho$, four scalars transform under Spin$(4)'$ as $(\mathbf{2}, \mathbf{2})$. This is the representation that corresponds to the tangent or cotangent bundle of $X$. Once $X$ is given a Riemannian metric, the two are equivalent; it will be more natural in what follows to think in terms of the cotangent bundle $T^*X$. The other two scalars are Spin$(4)'$ singlets but charged under $U(1)_R$. The geometrical picture is that $Y = T^*X \times \mathbb{R}^2$, where $X$ is embedded as the zero-section of $T^*X$ times a point in $\mathbb{R}^2$, and $U(1)_R$ acts as the rotation group of $\mathbb{R}^2$.

In a few favorable cases, such as $X = \mathbb{CP}^2$, $T^*X$ carries a complete Calabi-Yau metric, such that the zero-section is a Lagrangian submanifold. This puts the supersymmetry of the twisted model in a standard framework. Even when that is not so, $T^*X$ is a symplectic manifold, and a brane supported on its zero section is a Lagrangian brane in the $A$-model of $T^*X$, still giving a standard framework for the supersymmetry of the twisted model. (We expect that the analog of this for cases A$''$ and C$''$ is that $\Omega_2^+(X)$ or $S_+(X)$ will always carry a possibly unintegrable $G_2$ structure or Spin(7) structure, and that this suffices for topological applications. This point of view has not been studied systematically.)

(C$''$) With the third choice of $\rho$, four scalars transform under Spin$'(2)_\ell \times$ Spin$'(2)_r \times U(2)_R$ as $(\mathbf{2}, \mathbf{1}, \mathbf{2})_0$, where the subscript refers to the charge under the $U(1)_R$ center of $SU(2)_R$. The other two transform as $(\mathbf{1}, \mathbf{1}, \mathbf{1})_{\pm 2}$. The geometrical meaning is as follows. Let $S_+(X)$ be the

positive chirality spin bundle of $X$, viewed as a real vector bundle of rank 4. Then $Y$ can be identified as $S_+(X) \times \mathbb{R}^2$, where $X$ is embedded as the zero-section of $S_+(X)$ times a point in $\mathbb{R}^2$. $SU(2)_R$ acts on the fiber of $S_+(X) \to X$, commuting with its structure group $\text{Spin}(4)'$, and $U(1)_R$ acts on $\mathbb{R}^2$ by rotations.

In a few favorable cases, $S_+(X)$ carries a complete metric of $\text{Spin}(7)$ holonomy, with the zero section as a coassociative submanifold, providing a standard framework for the supersymmetry of the twisted model. Even when such a complete metric does not exist, this provides a sufficient description for our application to gauge theory.

## 2.4  M-Theory Variant

Instead of considering D3-branes in Type IIB superstring theory, we can consider M5-branes in M-theory. Here we use the fact that M-theory on $T^2 \times Z$, for a two-torus $T^2$ and any $Z$, goes over, in the limit that the $T^2$ is small, to Type IIB on $S^1 \times Z$. In this process, an M5-brane on $T^2 \times X$ (where $X$ is any submanifold of $Z$) goes over to a D3-brane on $X$ times a point in $S^1$. Since strings or branes wrapped on $S^1$ will not be important in anything we say, we can here decompactify $S^1$ and replace it by a copy of $\mathbb{R}$. The complex structure of the torus, $\tau$, plays the role of the complex coupling constant in 4d.

The upshot of this is that the geometrical descriptions that were described earlier have M-theory variants:[7]

($A'''$) In the first example, we can consider M-theory on $T^2 \times \Omega_2^+(X) \times \mathbb{R}^2$ with M5-branes wrapped on $T^2 \times X$ times a point in $\mathbb{R}^2$.

($B'''$) In the second example, we can consider M-theory on $T^2 \times T^*X \times \mathbb{R}$ with M5-branes wrapped on $T^2 \times X$ times a point in $\mathbb{R}$.

($C'''$) In the third example, we can consider M-theory on $T^2 \times S_+(X) \times \mathbb{R}$ with M5-branes wrapped on $T^2 \times X$ times a point in $\mathbb{R}$.

In each of these cases, we have the option to make the $T^2$ larger or smaller than $X$. In the limit that $T^2$ is very small, we return to the Type IIB description via D3-branes wrapped on $X$. This in turn can be described in terms of the four-dimensional twisted versions of $\mathcal{N} = 4$ super Yang-Mills theory, as described above. In the opposite limit that $X$ is very small compared to $T^2$, we get a description in terms of a conformal field theory on $T^2$. We will explore both limits in this paper.

Parallel M5-branes, which we have used in this explanation, give a particular realization of the $(2, 0)$ superconformal field theory in six dimensions. Instead of talking about M5-branes wrapped on $T^2 \times X$, we could more generally talk about the $(2, 0)$ model on $T^2 \times X$. This formulation is more general as it encompasses all groups of $A - D - E$ type.

Consider in more detail the twist of type ($A'''$). The (uncompactified) 6d theory has $\text{Spin}(5)_R \times \text{Spin}(6)$ global symmetry, where the first factor is the R-symmetry and the second factor describes (local) rotations of the 6d spacetime. When the 6d theory put on a spacetime of the form $X \times \Sigma^2$, where $X$ is an oriented 4-manifold and $\Sigma^2$ is a Riemann surface, the second factor naturally breaks into $\text{Spin}(2) \times \text{Spin}(4) \subset \text{Spin}(6)$. The two factors correspond to local rotations on $\Sigma^2$ and $X$ respectively. The topological twist is then realized by identifying the $SU(2)_\ell$ subgroup of $\text{Spin}(4) \cong SU(2)_\ell \times SU(2)_r$ with $SU(2)_R \cong \text{Spin}(3)_R \subset \text{Spin}(5)_R$ embedded in the standard way.

In the M-theory setting, the 6d type $A_1$ theory describes the dynamics of a stack of 2 M5 branes, with center of mass degrees removed. Then 6d spacetime rotations and the R-symmetry can be both embedded into the group of 11d rotations: $\text{Spin}(6) \times \text{Spin}(5)_R \subset \text{Spin}(11)$ where $\text{Spin}(5)_R$ correspond to the rotations along the directions orthogonal to the worldvolume of the 5-branes. The topological twist can be then realized by the following geometric

---

[7]In what follows, $X$ is short for the zero-section of $\Omega_2^+(X)$, $T^*X$, or $S_+(X)$.

background in M-theory:

$$
\begin{array}{llll}
\text{M-theory:} & \Omega_2^+(X) & \times\Sigma^2\times\mathbb{R}^2 \\
\text{5-branes:} & X & \times\Sigma^2
\end{array}
, \tag{4}
$$

where $\Omega_2^+(X)$ is the total space of the rank 3 vector bundle of the self-dual 2-forms over $X$. This construction follows from the fact that antisymmetric rank 2 tensors of $SO(4) \equiv \mathrm{Spin}(4)/\mathbb{Z}_2$ transforms as a triplet of $SU(2)_r \subset \mathrm{Spin}(4)$. After the topological twist $SU(2)_r$ is identified with the $\mathrm{Spin}(3)_R \subset \mathrm{Spin}(5)_R$ subgroup of the R-symmetry that corresponds to the rotations of the fibers of the normal bundle to the worldvolume of the 5-branes. The total space $\Omega_2^+(X)$ is a local $G_2$-manifold and $X$ is a coassociative cycle.

When $X$ is Kähler, as in the case of $X = \mathbb{CP}^2$, its holonomy is reduced to $U(2) \subset SO(4) \equiv \mathrm{Spin}(4)/\mathbb{Z}_2$. In particular, $SU(2)_\ell$ is reduced to its maximal torus $U(1)_\ell \subset SU(2)_\ell$. After the topological twist, this maximal torus is identified with the subgroup $U(1)_R \equiv \mathrm{Spin}(2)_R \subset \mathrm{Spin}(5)_R$ embedded in the standard way (i.e. as a subgroup corresponding to the rotations among 2 out of 5 normal directions). The three-dimensional real representation of $\mathrm{Spin}(3)_R$ decomposes into a complex 1-dimensional representation of $U(1)_R$ of charge 2 plus a trivial 1-dimensional real representation. Geometrically this correponds to the splitting of the rank 3 real vector bundle into a real rank 1 trivial bundle and a rank 1 complex vector bundle: $\Omega_2^+(X) = \mathbb{R} \times KX$ where $KX := \Lambda^2 T_{\mathbb{C}}^* X$ is the canonical bundle of $X$, considered as a complex manifold. The total space $KX$ of this canonical bundle is a local Calabi-Yau 3-fold.

## 2.5 Some Background Concerning the Anomaly

In cases A and C, one finds that if $S_{4\mathrm{d}}$ is the action of the theory, then the antiholomorphic dependence of $S_{4\mathrm{d}}$ on the gauge theory coupling parameter $\tau$ is $Q$-exact, meaning that

$$
\frac{\partial S_{4\mathrm{d}}}{\partial \bar{\tau}} = \{Q, \Lambda\}, \tag{5}
$$

for some functional $\Lambda$. (The details of case B are more complicated and depend on the choice of the parameter $t$.) Formally, it follows that as long as we only discuss $Q$-invariant operators and states, all computations in cases A or C will give results holomorphic in $\tau$. In fact, as already mentioned, the proof of decoupling of $Q$-exact operators depends on integration by parts and there is a possibility of an anomaly coming from a surface term at infinity. As explained in the introduction, from explicitly known formulas it appears that there is such a holomorphic anomaly in case A provided that $b_2^+(X) = 1$. In the present paper, we will aim to elucidate this anomaly.

We will primarily aim to understand the case of gauge group $SU(2)$ or $SO(3)$, which can be realized by a system of two D3-branes or two M5-branes after removing the center of mass degree of freedom. A possible anomaly will come from the Coulomb branch, which in the geometric description is the region in which two branes are widely separated in the *untwisted* directions. For example, for D3-branes on $\Omega_2^+(X)\times\mathbb{R}^3$, $T^*X\times\mathbb{R}^2$, or $S_+(X)\times\mathbb{R}^2$, the Coulomb branch is the region in which the D3-branes are widely separated in $\mathbb{R}^3$, $\mathbb{R}^2$, or $\mathbb{R}^2$. The overall center of mass motion of the D3-brane system is described by a free system. The Coulomb branch that we are interested in parametrizes the relative motion between the two D3-branes. So we can effectively describe the Coulomb branch asymptotically in terms of the degrees of freedom of a single D3-brane wrapped on $X$ but near infinity in the second factor of $\Omega_2^+(X)\times\mathbb{R}^3/\mathbb{Z}_2$, $T^*X\times\mathbb{R}^2/\mathbb{Z}_2$, or $S_+(X)\times\mathbb{R}^2/\mathbb{Z}_2$. (Here in describing the relative motion of the two D3-branes, we divide by $\mathbb{Z}_2$ to account for the Weyl group that exchanges the two D3-branes.) In the M-theory description, the story is similar. The Coulomb branch for the relative

motion of a pair of M5-branes can be described in terms of a single M5-brane wrapped on $T^2 \times X$ times a point near infinity in the last factor of $T^2 \times \Omega_2^+(X) \times \mathbb{R}^2/\mathbb{Z}_2$, $T^2 \times T^*X \times \mathbb{R}/\mathbb{Z}_2$, or $T^2 \times S_+(X) \times \mathbb{R}/\mathbb{Z}_2$.

We will find the anomaly by a careful study of the effective field theory on the Coulomb branch. In doing this, as remarked in the introduction, it is necessary to include in the theory on the Coulomb branch a certain half-BPS interaction that was originally described in [16] in the context of supersymmetric Yang-Mills theory, or its M-theory analog, which was originally described in [17, 18].

Formally, in any of the three twisted theories under discussion, one can calculate by a procedure of supersymmetric localization. Formally, one can show that modulo $Q$-exact terms, the path integral of any of the three twisted theories can be localized on configurations that satisfy $\{Q, \chi\} = 0$, where $\chi$ can be any of the fermion fields of the theory. The equations $\{Q, \chi\} = 0$ become a system of elliptic differential equations modulo the gauge group for bosonic fields in the theory. In theories A and B, these equations have been discussed in detail in [1] and [29], respectively. In theory C, the localization equations are similar to the "monopole" equations studied in [31]. Actually, the details of this localization procedure will not be our focus in the present paper; what we will be interested in here is precisely how this localization can fail. Since localization holds modulo $Q$-exact terms, the localization procedure will give an incomplete result if there is an anomaly at infinity on the Coulomb branch. The anomaly that we will explore violates the predictions of the formal localization procedure which implies holomorphy.

# 3 Holomorphic Anomaly in Four Dimensions

We begin in Section 3.1 by recalling a few facts about the holomorphic anomaly encountered [1] in the computation of the twisted partition function of supersymmetric Yang-Mills theory with gauge group $SO(3)$ on $\mathbb{CP}^2$. Our goal is to better understand how the anomaly relates to mock modularity and eventually to noncompactness of the Coulomb branch. In Section 3.2 we review the relevant facts about the effective action of the four-dimensional $\mathcal{N} = 4$ theory on the Coulomb branch. In Section 3.3 we give the derivation of the holomorphic anomaly from the path integral of the effective four-dimensional theory.

## 3.1 Mock Modularity of $\mathbb{CP}^2$ Partition Function

Since $H^2(\mathbb{CP}^2, \mathbb{Z}_2) \cong \mathbb{Z}_2$, an $SO(3)$ bundle $E \to \mathbb{C}^2$ has two possible values of $v = w_2(E)$: $v = 0$ with $v^2 = 0$ and $v \neq 0$ with $v^2 = -1$ modulo 4. Accordingly, there are two partition functions, which we denote by $Z_0(\tau)$ and $Z_1(\tau)$. It was shown in [1] using the work of Klyachko and Yoshioka [2, 3] that the partition functions can be expressed in terms of the Hurwitz-Kronecker class numbers denoted by $H(N)$ for $N \in \mathbb{Z}$. These class numbers are defined for $N > 0$ as the number of $PSL(2, \mathbb{Z})$-equivalence classes of integral binary quadratic forms of discriminant $-N$, weighted by the reciprocal of the number of their automorphisms (if $-N$ is the discriminant of an imaginary quadratic field $K$ other than $\mathbb{Q}(i)$ or $\mathbb{Q}(\sqrt{-3})$, this is just the class number of $K$), and for other values of $N$ by $H(0) = -1/12$ and $H(N) = 0$ for $N < 0$. These numbers vanish unless $N$ is 0 or $-1$ modulo 4.

With these definitions, the partition functions of the topologically twisted supersymmetric

$SO(3)$ Yang-Mills theory on $\mathbb{CP}^2$ [1] are given by

$$Z_0(\tau) \;=\; \frac{3}{\eta^3(\tau)}\widehat{h}_0(\tau), \tag{6}$$

$$Z_1(\tau) \;=\; \frac{3}{\eta^3(\tau)}\widehat{h}_1(\tau), \tag{7}$$

where $\eta(\tau)$ is the Dedekind eta function and

$$\widehat{h}_0(\tau) \;=\; \sum_{n\geq 0}H(4n)q^n + 2\tau_2^{-1/2}\sum_{n\in Z}\beta(4\pi n^2\tau_2)q^{-n^2},$$

$$\widehat{h}_1(\tau) \;=\; \sum_{n>0}H(4n-1)q^{n-\frac{1}{4}} + 2\tau_2^{-1/2}\sum_{n\in Z}\beta(4\pi(n+\tfrac{1}{2})^2\tau_2)q^{-(n+\frac{1}{2})^2}, \tag{8}$$

with $\tau = \tau_1 + i\tau_2$ and

$$\beta(t) = \frac{1}{16\pi}\int_1^\infty u^{-3/2}\exp(-ut)\,du\,, \tag{9}$$

which equals the complementary error function up to normalization.

Note that in [1,3] there is a factor of $\eta(\tau)^6$ in the denominator instead of $\eta(\tau)^3$ as in (6). The generating functions considered there are for compactified moduli spaces of anti-self-dual $U(2)$ connection with fixed value of the first Chern class. This is because those formulas were obtained in algebro-geometric setting, where the moduli spaces are realized as the moduli spaces of stable rank two sheaves on $\mathbb{CP}^2$ with fixed first and second Chern classes. The extra $1/\eta(\tau)^3 = 1/\eta(\tau)^{\chi(\mathbb{CP}^2)}$ compared to the $SU(2)$ case can be understood as the contribution of the abelian point-like instantons from the diagonal $U(1)$ subgroup of $U(2)$. The formulas (7) are consistent with the fact that, by lifting the theory to the 6d (2,0) theory on $\mathbb{CP}^2 \times T^2$, one can argue that the physically normalized partition functions $Z_\nu(\tau)$ should transform as a vector valued modular form of weight *zero*, with a multiplier system determined by 't Hooft anomalies. The case of $U(2)$ gauge group will be discussed further in Appendix B.

The functions $\{\widehat{h}_\ell(\tau)\}$ are not purely holomorphic because of the second term in (8) and satisfy the *holomorphic anomaly equation*:

$$\tau_2^{3/2}\frac{\partial}{\partial\bar\tau}\widehat{h}_0 \;=\; \frac{1}{16\pi i}\sum_{n\in\mathbb{Z}}\bar q^{n^2}\,, \tag{10}$$

$$\tau_2^{3/2}\frac{\partial}{\partial\bar\tau}\widehat{h}_1 \;=\; \frac{1}{16\pi i}\sum_{n\in\mathbb{Z}}\bar q^{(n+\frac{1}{2})^2}\,. \tag{11}$$

A nontrivial fact [25] is that $\widehat{h}(\tau) = \begin{pmatrix}\widehat{h}_0(\tau)\\\widehat{h}_1(\tau)\end{pmatrix}$ transforms as a vector valued modular form with weight 3/2 under the modular group $\Gamma_0(4)$. In particular,

$$\begin{pmatrix}\widehat{h}_0(-1/\tau)\\\widehat{h}_1(-1/\tau)\end{pmatrix} = \left(\frac{\tau}{i}\right)^{3/2}\cdot\frac{-1}{\sqrt{2}}\begin{pmatrix}1 & 1\\1 & -1\end{pmatrix}\begin{pmatrix}\widehat{h}_0(\tau)\\\widehat{h}_1(\tau)\end{pmatrix}. \tag{12}$$

Since $\widehat{h}_0$ is invariant under $T$ and $\widehat{h}_1$ under $T^4$, $\widehat{h}_0$ has the expected modular transformation law under the group $\Gamma_0(4)$ generated by $T$ and $ST^4S$, as expected from S-duality [1].

The holomorphic parts of $\{\widehat{h}_\ell(\tau)\}$ are

$$h_0(\tau) \;=\; \sum_{n=0}^\infty H(4n)q^n\,, \tag{13}$$

$$h_1(\tau) \;=\; \sum_{n=1}^\infty H(4n-1)q^{n-\frac{1}{4}}\,. \tag{14}$$

In modern terminology [26–28], $h(\tau)$ is a *vector-valued pure mock modular form* with holomorphic *shadow* $g(\tau)$ with components

$$g_0(\tau) = c\sum_{n\in\mathbb{Z}} q^{n^2}, \tag{15}$$

$$g_1(\tau) = c\sum_{n\in\mathbb{Z}} q^{(n+\frac{1}{2})^2}, \tag{16}$$

where $c = \frac{1}{16\pi i}$ is the overall normalization for which there is no standard convention. The vector $h(\tau)$ is holomorphic but not modular. Addition of the correction terms in (8) constructed from the shadow vector $g(\tau)$ yields its *modular completion* $\widehat{h}(\tau)$. The completion is modular but not holomorphic. This incompatibility between modularity and holomorphy is the essence of mock modularity.

The connection to mock modularity can be seen more simply by combining the two components of the vector-valued mock modular form into a single mock modular form defined by $\mathbf{H}(\tau) = h_0(4\tau) + h_1(4\tau)$. This gives the Zagier mock modular form [25] which is the generating function for the Hurwitz-Kronecker class numbers:

$$\mathbf{H}(\tau) := \sum_{N=0}^{\infty} H(N) q^N = -\frac{1}{12} + \frac{1}{3}q^3 + \frac{1}{2}q^4 + q^7 + q^8 + q^{11} + \cdots \tag{17}$$

It is a 'pure mock modular form' of weight $3/2$ on $\Gamma_0(4)$ and shadow the classical theta function $\vartheta(\tau) = \sum q^{n^2}$. See [27,28] for the definition and further discussion. We see from the $q$-expansion of $\mathbf{H}(\tau)$ that it has no poles at $q = 0$ and hence is strongly holomorphic. Consequently, its Fourier coefficients grow very slowly. This is exceptional. In fact, up to minor variations the Zagier mock modular form is essentially the *only* known non-trivial example of a strongly holomorphic pure mock modular form.

The components $\{h_\ell\}$ are most naturally regarded as the vector of theta coefficients of a mock Jacobi form $\mathcal{H}(\tau, z)$ defined by

$$\mathcal{H}(\tau, z) := h_0(\tau)\vartheta_{1,0}(\tau, z) + h_1(\tau)\vartheta_{1,1}(\tau, z) = \sum_{\substack{n,\, r \in \mathbb{Z} \\ 4n-r^2 \geq 0}} H(4n - r^2) q^n y^r, \tag{18}$$

where $y = e^{2\pi i z}$ and

$$\vartheta_{m,\ell}(\tau, z) := \sum_{\substack{r \in \mathbb{Z} \\ r \equiv \ell \,(\mathrm{mod}\, 2m)}} q^{r^2/4m} y^r \qquad (\ell \bmod 2m) \tag{19}$$

are the level $m$ theta functions. Following the definition in [28], one can check that $\mathcal{H}(\tau, z)$ is a mock Jacobi form of weight 2 and index 1 with holomorphic anomaly (up to normalization)

$$\overline{\vartheta_{1,0}(\tau, 0)}\,\vartheta_{1,0}(\tau, z) + \overline{\vartheta_{1,1}(\tau, 0)}\,\vartheta_{1,1}(\tau, z). \tag{20}$$

Note that a similar relation between components $\{h_\ell\}$ form and a single Jacobi mock modular form appears in the decomposition of the Vafa-Witten $U(2)$ partition functions into the sum of products of $SO(3)$ and $U(1)$ partition functions. However, the latter decomposition is different because the coefficients in that case are *anti-holomorphic* theta functions. We will discuss it in more detail in Appendix B.

## 3.2 Wess-Zumino Term in Four Dimensions

As explained in the introduction, the holomorphic anomaly of interest is given by a contribution from the boundary of the space of the bosonic zero-modes. Saturation of fermionic zero-modes

in this region is governed by the two-fermion superpartner of a Wess-Zumino term which we describe below and in Appendix A.

Consider a four-dimensional $\mathcal{N} = 4$ super Yang-Mills theory with gauge group $G$ spontaneously broken to $H \times U(1)$ by the vacuum expectation value of the scalar field $\Phi$ valued in $\mathbb{R}^6$. The bosonic part of the effective action, after coupling to background gauge fields of $SO(6)_R$, contains the following Wess-Zumino term [18, 32, 33]:

$$S_{\text{4d WZ}} = 2\pi i \, \frac{n_W}{2} \int_{\Xi^5} \eta_5 \tag{21}$$

with

$$\eta_5 := \frac{1}{120\pi^3} \epsilon_{I_1 I_2 I_3 I_4 I_5 I_6} [(D_{i_1}\hat{\Phi})^{I_1} (D_{i_2}\hat{\Phi})^{I_2} (D_{i_3}\hat{\Phi})^{I_3} (D_{i_4}\hat{\Phi})^{I_4} (D_{i_5}\hat{\Phi})^{I_5}$$
$$+ \frac{5}{2} F^{I_1 I_2}_{i_1 i_2} (D_{i_3}\hat{\Phi})^{I_3} (D_{i_4}\hat{\Phi})^{I_4} (D_{i_5}\hat{\Phi})^{I_5} + \frac{15}{4} F^{I_1 I_2}_{i_1 i_2} F^{I_3 I_4}_{i_3 i_4} (D_{i_5}\hat{\Phi})^{I_5}] \hat{\Phi}^{I_6} dx^{i_1} \wedge dx^{i_2} \wedge dx^{i_3} \wedge dx^{i_4} \wedge dx^{i_5}, \tag{22}$$

where $\Phi^I, I = 0, \ldots, 5$ are the six scalar fields of the unbroken $U(1)$, $\hat{\Phi}^I := \Phi^I / \|\Phi\|$, $\|\Phi\|^2 := \Phi^I \Phi^I$, $(D_i \Phi)^I := \partial_i \Phi^I - A_i^{IJ} \Phi^J$, $A$ is the background $SO(6)_R$ connection and $F$ is its curvature. As usual, the integral is over a five-dimensional manifold $\Xi^5$ which has $X$ as its boundary. The factor of $i$ is due to the fact that we work in Euclidean spacetime (here and in the rest of the paper we use the convention in which the integrand of the path integral is $e^{-S}$). This term compensates for the deficit in the 't Hooft anomaly of the $SU(4)_R$ R-symmetry.[8] The number $n_W = \dim G - \dim H - 1$ is the number of 'W-bosons.' In the special case when $G = SO(3) = SU(2)/\mathbb{Z}_2$ and $H = 1$, one obtains $n_W = 2$.

We can label fields such that in the description via M5-branes wrapped on $\mathbb{R}^4 \times T^2$, $\Phi^0$ is compact and $\Phi^I, 1 \le I \le 5$ are noncompact fields that correspond to oscillations of the M5-branes in the transverse $\mathbb{R}^5$. Only the $Spin(5)_R$ symmetry acting on the transverse oscillations is manifest in this description; it is enhanced to $Spin(6)_R$ in the limit of small $T^2$ as $\Phi^0$ decompactifies. The relation between the topological terms in six and four dimensions will be discussed in more detail in Section 4.1.

Geometrically, $\eta_5 = \hat{\Phi}^*(e_5)$ is the pull-back of the global Euler angular form $e_5$ on the total space of an $S^5$ bundle $\mathcal{E} \to \Xi^5$ to the base space $\Xi^5$ by the section $\hat{\Phi} : \Xi^5 \to \mathcal{E}$. In general, for an $S^n$ sphere bundle $\pi : \mathcal{E} \to \Xi$, the fiber can be thought of as a sphere in the fiber $\mathbb{R}^{n+1}$ of a real vector bundle $V$. One can define the global Euler angular form $e_n \in \Omega^n(\mathcal{E})$ with the following properties. Its restriction to the fiber gives the volume form $\omega_n$ normalized such that $\int_{S^n} \omega_n = 1$. Moreover, $de_n = -\pi^*(e(V))$ where $e(V)$ is the standard representative of the Euler class of the bundle $V \to \Xi$. For odd $n$, the form $e_n$ is not closed in general when the bundle is non-trivial, as in the case above for $n = 5$. For even $n$, the Euler density form $e(V)$ is identically zero and $e_n$ is closed; this fact will be used in Section 4 for $n = 4$. Its de Rham cohomology class satisfies $[e_n]^2 = \pi^*(p_{n/2}(V))$, where $p_{n/2}(V)$ is the $n/2$-th Pontryagin class of $V$. See, for example, [34] for details and explicit general formulas for $e_n$.

As shown in [35], the Wess-Zumino term is part of the $\mathcal{N} = 4$ completion of the Dine-Seiberg term [16]. Restricted to the $\mathcal{N} = 2$ vector multiplet, the Dine-Seiberg term is given by a logarithmic prepotential. However, to compute the holomorphic anomaly, we will need not only the vector multiplet couplings but also the couplings involving both the $\mathcal{N} = 2$ vector multiplet and the hypermultiplets including the auxiliary fields. These are related by supersymmetry to the second term in (22) which is linear in the R-symmetry curvature $F$. The supersymmetrization is therefore more elaborate than what is available in the literature and will be discussed in Appendix A.

---

[8]The anomaly in question corresponds to the term proportional to the third Chern class $\text{Tr}F^3/24\pi^3$ of the corresponding R-symmetry bundle in the degree six anomaly polynomial.

### 3.3 Holomorphic Anomaly

To compute the holomorpic anomaly from the gauge theory path integral, we first represent the full four-dimensional effective action $S_{4d}$ as

$$\frac{\partial S_{4d}}{\partial \bar{\tau}} = \{Q, \Lambda\}. \tag{23}$$

A formula like (23) holds both microscopically and (therefore) also in the Coulomb branch effective field theory. We want this formula in the Coulomb branch effective field theory, in which we will do the localization computation, and in a formalism in which the scalar supercharges which are used in the localization are realized off-shell. An off-shell realization of the scalar supercharges gives a precise framework for the localization computation. The off-shell realization of the scalar supercharges in the topologically twisted $\mathcal{N} = 4$ theory was considered in [1,36,37].

The effective action on the Coulomb branch is the sum of a quadratic action, a half-BPS coupling that is the supersymmetric completion of the Wess-Zumino coupling of eqn. (21), and various couplings of higher order that are in large supermultiplets. These higher order couplings have no BPS properties, and we do not expect them to be relevant. The Wess-Zumino coupling does not depend on $\tau$ or $\bar{\tau}$, and, in a formalism in which the scalar supercharges are realized off-shell, the same is true for its completion that is invariant under those supercharges. So the half-BPS coupling on the Coulomb branch will not contribute to $\Lambda$. Thus, we can evaluate $\Lambda$ just using quadratic the quadratic part $S_{4d}^{\text{free}}$ on the Coulomb branch effective action $S_{4d}$:

$$\frac{\partial S_{4d}}{\partial \bar{\tau}} = \frac{\partial S_{4d}^{\text{free}}}{\partial \bar{\tau}} = \{Q, \Lambda\}. \tag{24}$$

To determine the anomaly, we will be interested in the integral over the zero-modes of the path integral near the boundary. The zero-modes can be readily determined. In the twisted theory on $\mathbb{CP}^2$, the six real scalars of the untwisted theory turn into four real scalars and a complex section of the canonical bundle. Since there are no harmonic sections of the canonical bundle over $\mathbb{CP}^2$, there are only four real bosonic zero-modes corresponding to the four scalars. Since $b_1 = 0$, the gauge field in the Coulomb branch effective action has no zero-modes. Finally, the auxiliary fields in the twisted theory are a self-dual 2-form $H^{(2+)}$ and a 1-form $\tilde{H}^{(1)}$. Since $b_1 = 0$ and $b_2^+ = 1$, one obtains a single zero-mode of the auxiliary field, which we discuss in more detail in Appendix A. The fermions in the twisted theory consist of two scalars, two self-dual 2-forms and two vectors. Since $b_1 = 0$ and $b_2^+ = 1$ one obtains four fermionic zero-modes. In summary, there are four bosonic zero-modes of the four scalars, a single zero-mode of the bosonic auxiliary field, and four fermionic zero-modes. It may seem unusual in a supersymmetric theory that we have 1 extra bosonic degree of freedom, but we have to divide by the volume of the gauge group and hence the (compact) zero-mode of the gauge parameter counts as $-1$ bosonic degrees of freedom.

Since $\mathbb{CP}^2$ is Kähler, a $Spin(4) = SU(2) \times SU(2)$ subgroup of $SU(4)_R$ R-symmetry remains unbroken after twisting. There are four scalar supercharges $Q_A$ and $Q_{\dot{A}}$, transforming as $(\mathbf{2}, \mathbf{1}) \oplus (\mathbf{1}, \mathbf{2})$ respectively. The four fermion zero-modes transform likewise,[9] and we denote them by $\chi^A$ and $\bar{\chi}^{\dot{A}}$. The four bosonic zero-modes transform as $(\mathbf{2}, \mathbf{2})$ and we denote them by $u_{A\dot{A}}$ with the reality conditions $(u^{A\dot{A}})^* = \epsilon_{AB}\epsilon_{\dot{A}\dot{B}}u^{B\dot{B}}$. The zero mode of the auxiliary field transforms as a scalar and we denote it as $\mathcal{H}$.

Restricted to the zero-modes, the four off-shell supercharges described in Appendix A, in

---

[9]In our conventions, spinor indices are raised and lowered by $\chi_A = \epsilon_{AB}\chi^B$ with $\epsilon_{12} = -1$ and $\epsilon^{12} = +1$ and similarly for the dotted indices.

particular in (126), take the form

$$
\begin{aligned}
Q_A|_{\text{zm}} &= \mathcal{H}\frac{\partial}{\partial \chi^A} + \bar{\chi}^{\dot{A}}\frac{\partial}{\partial u^{A\dot{A}}} \\
\bar{Q}_{\dot{A}}|_{\text{zm}} &= \mathcal{H}\frac{\partial}{\partial \bar{\chi}^{\dot{A}}} - \chi^A \frac{\partial}{\partial u^{A\dot{A}}},
\end{aligned}
\tag{25}
$$

where zm stands for the zero-modes.

The effective action restricted to the zero-modes can have at most four fermions. The most general action consistent with the unbroken $Spin(4)_R$ symmetry takes the form

$$
S_{4\text{d}}|_{\text{zm}} = G(|u|^2) + K_{AB}(u)\chi^A\chi^B + K_{A\dot{B}}(u)\chi^A\bar{\chi}^{\dot{B}} + K_{\dot{A}\dot{B}}(u)\bar{\chi}^{\dot{A}}\bar{\chi}^{\dot{B}}
$$
$$
+ R(|u|^2)\epsilon_{AB}\epsilon_{\dot{A}\dot{B}}\chi^A\chi^B\bar{\chi}^{\dot{A}}\bar{\chi}^{\dot{B}}, \tag{26}
$$

where $|u|^2 := u^{A\dot{A}}u_{A\dot{A}}$. The coefficients depend also on $\mathcal{H}$ and the flux of the gauge field $n = \int_{\mathbb{CP}^1} F_A/2\pi$ (which for the gauge group $SO(3)$ is valued in $\frac{1}{2}\mathbb{Z}$), but we do not show this dependence explicitly. Demanding $Q_A S|_{\text{zm}} = Q_{\dot{A}}S|_{\text{zm}} = 0$, one obtains

$$
\begin{aligned}
K_{AB}(u) &= 0, & K_{A\dot{B}}(u) &= -\frac{2u_{A\dot{B}}G'(|u|^2)}{\mathcal{H}}, \\
K_{\dot{A}\dot{B}}(u) &= 0, & R(|u|^2) &= \frac{2G'(|u|^2) + |u|^2 G''(|u|^2)}{2\mathcal{H}^2}.
\end{aligned}
\tag{27}
$$

The action is thus completely determined in terms of a single function $G(|u|^2)$. Since we are interested in the action at the boundary corresponding to $|u|^2$ large, we consider the expansion

$$
G(|u|^2) = C_0(\mathcal{H}, n) + \frac{C_1(\mathcal{H}, n)}{|u|^2} + \frac{C_2(\mathcal{H}, n)}{|u|^4} + \dots \tag{28}
$$

The function $C_0(\mathcal{H}, n)$ is determined by the free part of the original action and hence is a homogeneous polynomial of degree two in $\mathcal{H}$ and $n$. The higher $C_k(\mathcal{H}, n)$ for $k > 0$ are determined by the interacting part of the action related by supersymmetry to the Wess-Zumino term. The Wess-Zumino term is scale invariant if all scalar fields scale with weight one. Hence, the $C_k(\mathcal{H}, n)$ are homogeneous polynomials of degree $2k$ in $\mathcal{H}$ and $n$. The contribution to the integral over the zero-modes from the boundary at infinity is determined by only the first two terms in (28) because the subsequent terms fall off rapidly for large $|u|$.

The function $C_0(\mathcal{H}, n)$ is determined in Appendix A and is given by

$$
S_{4\text{d}}^{\text{free}}|_{\text{zm}} = C_0(\mathcal{H}, n) = -\frac{\tau_2}{\pi}\mathcal{H}(\mathcal{H} - 4\pi n) + 2\pi i \tau n^2. \tag{29}
$$

The function $C_1(\mathcal{H}, n)$ is a homogeneous polynomial of degree two and hence there are only three possible terms. The condition that $K_{A\dot{B}}$ does not have $\mathcal{H}$ in the denominator implies that $C_1(\mathcal{H}, n)$ has no term constant in $\mathcal{H}$. We note that according to (27), for $G(|u|^2) = 1/|u|^2$ we obtain $R(|u|^2) = 0$ which puts no constraint on the term linear in $\mathcal{H}$. In summary, $C_1(\mathcal{H}, n)$ takes the general form

$$
C_1(\mathcal{H}, n) = i\, a\mathcal{H}(2\pi n + b\,\mathcal{H}) \tag{30}
$$

for some numerical constants $a$ and $b$ that need to determined.

The term linear in $\mathcal{H}$ can be related by supersymmetry to the Wess-Zumino term using the off-shell formalism realizing four scalar supercharges in the twisted theory as described in Appendix A. One obtains $a = 3/\pi$ from the relevant bosonic terms. Note that we have introduced the overall factor of $i$ in (30) for convenience because the Wess-Zumino term is imaginary in Euclidean action. The term quadratic in $\mathcal{H}^2$ cannot be related by supersymmetry to the Wess-Zumino term using only the four off-shell scalar supercharges described in the

Appendix. However, the constant $b$ can be determined by imposing $Spin(6)_R$ R-symmetry and using an off-shell realization of eight supercharges in the $\mathcal{N} = 2$ supergravity formalism in the untwisted theory. One obtains $b = -1$ from the relevant bosonic terms.

The nonholomorphic variation of the action (24) for $Q = Q_1 + \bar{Q}_1$ is given by $S_{4d}^{\text{free}}|_{\text{zm}} = \{Q|_{\text{zm}}, \Lambda_{\text{zm}}\}$ with

$$\Lambda|_{\text{zm}} = \frac{-1}{4\pi i}(\mathcal{H} - 4\pi n)(\chi^1 + \bar{\chi}^1), \tag{31}$$

as explained in more detail in Appendix A. The holomorphic anomaly then takes the form

$$\frac{\partial Z_\nu}{\partial \bar{\tau}} = \langle \{Q, \Lambda\} \rangle = \frac{1}{\eta(\tau)^3} \sum_{n \in \mathbb{Z} + \nu/2} C_{\text{zm}}(n), \tag{32}$$

where the terms on the right have the following origin.

- The factor $\eta^{-3} = \eta^{-\chi(\mathbb{CP}^2)}$ arises from $U(1)$ point-like instantons (see details below).
- The sum is over integral $U(1)$ fluxes with $\nu = w_2(E) \cong \mathbb{Z}_2$ being the discrete flux.
- The contributions from nonzero modes cancels due to supersymmetry as usual, and the coefficient $C_{\text{zm}}(n)$ is given by a *finite-dimensional* integral over the zero-modes:

$$C_{\text{zm}}(n) = C \int d^4 u \, d^4 \chi \, d\mathcal{H} \, Q|_{\text{zm}} (\Lambda|_{\text{zm}} e^{-S_{4d}|_{\text{zm}}}), \tag{33}$$

where as determined above

$$S_{4d}|_{\text{zm}} = -\frac{\tau_2}{\pi} \mathcal{H}(\mathcal{H} - 4\pi n) - 2\pi i \tau n^2 + 2i \, a(2\pi n + b\mathcal{H}) u_{A\dot{B}} \chi^A \bar{\chi}^{\dot{B}}/|u|^4 + \dots \tag{34}$$

and $C$ is a numerical constant that depends on the normalization of the path integral measure. This expressions are of course valid only away from the origin $u = 0$, where the description in terms of abelian gauge theory breaks down and extra massless degrees of freedom appear. This does not affect the calculation because the integral over $u$ is eventually reduced to an integral over the boundary at infinity using Stokes's theorem. There is no contribution to the holomorphic anomaly from a vicinity of $u = 0$ as this point is away from the boundary of the field space of the original non-abelian gauge theory.

We recall that a single point-like instanton can be formally understood as a singular configurations of the gauge field with the instanton charge (i.e. the second Chern class $c_2$) concentrated at a single point. For a non-abelian gauge field they can arise as limits of smooth field confugurations. It is known that in order to have an agreement with other descriptions of the gauge theory (including string theory), such point-like instantons should also be included in the abelian case. They also arise naturally when one considers a non-commutative deformation of the space-time [38] and in the algebro-geometric setting, where $U(1)$ instantons are interpreted as semistable rank 1 sheaves. The moduli space of point-like instantons on $X$ with a total instanton charge $k$ is the Hilbert scheme of $k$ points of $X$ (which is a resolution of a $k$-th symmetric product $\text{Sym}^k X$). Their total contribution to the Vafa-Witten partition function is to multiply it by a factor $\eta(\tau)^{-\chi(X)}$ [1, 39, 40].

As was explained in the introduction, in the 4d setup, to determine the absolute normalization of the path integral and therefore the constant $C$ in (33), one needs to compare the normalization used at short distances to put the holomorphic expansion in the form (1) with the normalization of the path integral measure in the low energy effective field theory on the Coulomb branch. This normalization can be affected, among other things, by topological terms

– linear combinations of the Euler characteristic and signature of $X$ – which might appear in the effective action on the Coulomb branch after integrating out massive modes. By contrast, in the 2d setup of Section 4, the relevant path integral can be interpreted as a Hilbert space trace; this provides a direct way to determine its normalization. Both 4d and 2d descriptions originate from the underlying 6d theory on $X \times T^2$, where the partition function also has interpretation as a Hilbert space trace. However, once the $T^2$ is forgotten, fixing the normalization becomes more involved.

The integrals over two out of four fermionic zero-modes in (33) are saturated by $Q|_{\text{zm}}$ and $\Lambda|_{\text{zm}}$. The other two should be saturated by terms in the action (26) that are quadratic in fermionic zero-modes. Bringing down the fermionic terms from the exponential, performing the Grassmann integrals, and using Stokes's theorem, one obtains

$$C_{\text{zm}}(n) = \frac{a\,C}{4\pi} \int d\mathcal{H}\,(\mathcal{H} - 4\pi n)(b\,\mathcal{H} + 2\pi n) e^{\frac{\tau_2}{\pi}(\mathcal{H} - 2\pi n)^2 - 2\pi i \bar{\tau} n^2} \cdot \int_{S^3} (\zeta_3 + \tilde{\zeta}_3), \qquad (35)$$

where

$$\zeta_3 := \frac{(u^{11}du^{12} - u^{12}du^{11})du^{22}du^{21}}{|u|^4}, \qquad (36)$$

$$\tilde{\zeta}_3 := \frac{(u^{21}du^{11} - u^{11}du^{21})du^{12}du^{22}}{|u|^4}, \qquad (37)$$

and we have divided the integral by two to take into account the quotient by the $\mathbb{Z}_2$ Weyl group on $S^3$. We show in the appendix that the integral over $\zeta_3 + \tilde{\zeta}_3$ gives $2\pi^2$. The Gaussian integral over $\mathcal{H}$ can be readily performed to obtain

$$C_{\text{zm}}(n) = \frac{-i\pi^3 aC}{4} \left( \tau_2^{-3/2} - 4i(1+b)\frac{\partial}{\partial \bar{\tau}} \tau_2^{-1/2} \right) \bar{q}^{n^2}. \qquad (38)$$

Note that with our choice of normalization of the auxiliary field $\mathcal{H}$ the corresponding quadratic term has a "wrong" sign in the exponential. To make it convergent, the integral over $\mathcal{H}$ is performed along the imaginary axis. This is the origin of the factor of $i$ in the result of the integration. The overall sign is somewhat ambiguous and depends on the choice of orientation in the space of zero modes. We do not address the question of fixing it in this paper. Combining (32) and (38) and plugging in $a = \frac{3}{\pi}$, $b = -1$, one obtains

$$\frac{\partial Z_\nu}{\partial \bar{\tau}} = \frac{3\pi C}{4i\,\tau_2^{3/2}\eta(\tau)^3} \sum_{n \in \mathbb{Z}} \bar{q}^{(n+\nu/2)^2} \qquad (39)$$

in agreement with the expected formulas (10) and (11) if $C = (2\pi)^{-2}$ in (38).

# 4 Holomorphic Anomaly in Two Dimensions

The non-compactness of the target space of the two-dimensional sigma model obtained by dimensionally reducing the six-dimensional type $A_1$ theory on $\mathbb{CP}^2$ can lead to a holomorphic anomaly. This anomaly can receive a contribution only from the boundary of field space, so it suffices to determine the sigma model in this region. The noncompact bosonic zero-modes of the $A_1$ theory parametrize the separation between a pair of M5-branes. When these fields have large expectation values, the six-dimensional theory can be approximated by a single $(2,0)$ tensor multiplet valued in the Cartan subalgebra $\mathfrak{u}(1) \subset \mathfrak{su}(2)$. This is the regime in which we work.

We start by reviewing relevant facts about the effective action of the six-dimensional $(2,0)$ theory on the tensor branch in Section 4.1. Then, in Section 4.2 we determine the effective two-dimensional theory obtained by compactification of the six-dimensional theory on $\mathbb{CP}^2$. The two-dimensional theory is similar to a heterotic sigma-model. The Wess-Zumino-like terms in the 6d action lead to the Wess-Zumino terms in the 2d action. In Section 4.3 we review relevant facts about 2d $(0,1)$ supersymmetric nonlinear sigma models. As in four dimensions, the holomorphic anomaly is determined by the supersymmetrization of the Wess-Zumino term. In Section 4.4 we derive the holomorphic anomaly from the path integral of the two-dimensional theory.

## 4.1 Six-dimensional Effective Action

The $(2,0)$ theory in six dimensions is characterized by a choice of a Lie algebra $\mathfrak{g}$, which is assumed to be a direct sum of simply-laced Lie algebras and $\mathfrak{u}(1)$ factors. For each $\mathfrak{u}(1)$, the theory contains a $(2,0)$ abelian tensor multiplet which consists of a 2-form gauge field $B$ with self-dual 3-form field strength $(dB = *dB)$, five scalar fields $\Phi^a$ $(a = 1, \ldots, 5)$, transforming as a vector of $Spin(5)_R$, and fermionic fields in the $(\mathbf{4}, \mathbf{4})$ representation of $Spin(6) \times Spin(5)_R$ where $Spin(6)$ is the six dimensional rotation symmetry and $Spin(5)_R$ is the R-symmetry.

It was argued in [18] that 'higgsing' $\mathfrak{g} \to \mathfrak{h} \oplus \mathfrak{u}(1)$ of the $(2,0)$ model in six dimensions produces a Wess-Zumino-like term in the effective action on a six-dimensional space $M$, of the form

$$S_{6\text{d WZ}} = \frac{c(\mathfrak{g}) - c(\mathfrak{h})}{6} 2\pi i \int_{\Xi^7} \Omega_3 \wedge d\Omega_3, \tag{40}$$

where $c(\mathfrak{g}) = \dim \mathfrak{g} \cdot h_\mathfrak{g}^\vee$ and $\partial \Xi^7 = M$. This term compensates for the mismatch of the 't Hooft anomaly for the $SO(5)_R$ R-symmetry. The overall factor of $i$ is present because we consider Euclidean action. con The 3-form $\Omega_3$ is (locally) defined by descent:

$$d\Omega_3 := \eta_4 \tag{41}$$

with

$$
\begin{aligned}
\eta_4 := \frac{1}{64\pi^2} \epsilon_{a_1 a_2 a_3 a_4 a_5} \big[ & (D_{i_1}\hat{\Phi})^{a_1}(D_{i_2}\hat{\Phi})^{a_2}(D_{i_3}\hat{\Phi})^{a_3}(D_{i_4}\hat{\Phi})^{a_4} \\
& - 2F_{i_1 i_2}^{a_1 a_2}(D_{i_3}\hat{\Phi})^{a_3}(D_{i_4}\hat{\Phi})^{a_4} + F_{i_1 i_2}^{a_1 a_2}F_{i_3 i_4}^{a_3 a_4}\big]\hat{\Phi}^{a_5} dx^{i_1} \wedge dx^{i_2} \wedge dx^{i_3} \wedge dx^{i_4},
\end{aligned} \tag{42}
$$

where $\Phi^a$, $a = 1, \ldots, 5$ are the five scalar fields of the $\mathfrak{u}(1)$ tensor multiplet, $\hat{\Phi}^a := \Phi^a/\|\Phi\|$, $\|\Phi\|^2 := \Phi^a \Phi^a$, $(D_i\Phi)^a := \partial_i\Phi^a - A_i^{ab}\Phi^b$, $A$ is a background $SO(5)_R$ connection and $F$ is its curvature. For the term (40) to be well defined, it is necessary that $\eta_4$ is closed. This is true in general. We check it explicitly in Section C.2 when only an $Spin(2)$ subgroup of the $Spin(5)_R$ connection is turned on relevant for topological twisting on a Kähler 4-manifold. Geometrically, $\eta_4 = \hat{\Phi}^*(e_4)$ is the pull-back of the global Euler angular form $e_4$ on the total space of $S^4$ sphere bundle $\mathcal{E} \to M$ to the base space $M$ by the section $\hat{\Phi} : M \to \mathcal{E}$. See discussion in Section 3.2.

In [17, 18] it was argued that the effective action also contains a topological term that governs the coupling of the Skyrmionic string to the $B$-field:

$$S_{6\text{d Sk}} = \frac{i n_W}{2} \int_M B \wedge \eta_4 = \frac{i n_W}{2} \int_{\Xi^7} dB \wedge \eta_4, \tag{43}$$

where $n_W = \dim \mathfrak{g} - \dim \mathfrak{h} - 1$ is the number of $W$-bosons in the five-dimensional theory obtained by dimensional reduction, the same as the $n_W$ that appeared in the Section 3.2. In the formula above and later, the $B$-field is normalized such that large gauge transformations

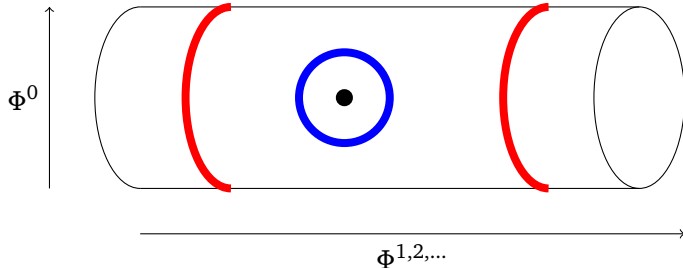

Figure 1: The 5-cycles $S^5$ (in blue) and $S^4 \times S^1$ (in red) are homologous in $\mathbb{R}^5 \times S^1$, the space where the scalar fields $\Phi^I$, $I = 0 \ldots 5$ are valued. This is equivalent to the statement that the flux through the boundary at infinity is preserved when the radius of $S^1$ is taken to be infinitely large.

shift it by an element of $2\pi H^2(M, \mathbb{Z})$. By reducing the 6d theory to 5d, one can relate (43) to a term linear in the gauge field found in [33] by a one-loop calculation, with the coefficient in agreement with the formula above. By reducing the 6d theory further to 4d, the term (43) can be related to the Wess-Zumino term (21), which was shown in Section 3.3 to be responsible for the holomorphic anomaly.

To see the relation between these two terms in more detail, and verify the consistency of the coefficients in front of the integrals, consider $\Xi^7 = T^2 \times \Xi^5$. Writing $\Phi^0 = \int_{T^2} B$, one obtains a compact boson valued in a circle. In the 4d limit its radius becomes infinitely large. In a trivial R-symmetry background, the action $S_{\text{6d Sk}}$ reduces to

$$\frac{in_W}{2} \int_{\Xi^5} d\Phi_0 \wedge \hat{\Phi}^*(\omega_4), \tag{44}$$

which *a priori* appears quite different from the integral of $\hat{\Phi}^*(\omega_5)$ in (21). However, both terms can be interpreted as having $n_W/2$ units of flux at the infinity of the space where the scalar fields $\Phi^I$, $I = 0, \ldots, 5$ are valued. In (44) this space is $\mathbb{R}^5 \times S^1$ with boundary at infinity $S^4 \times S^1$. In the 4d limit the size of the 2-torus $T^2$ is taken to zero, the radius of $S^1$ becomes infinite and the space where the scalar fields are valued becomes $\mathbb{R}^6$ with boundary at infinity $S^5$. This deformation is shown in Figure 1.

The two six-dimensional terms above can be combined into a compact expression

$$S_{\text{6d Sk}} + S_{\text{6d WZ}} = \frac{in_W}{2} \int_{\Xi^7} H_3 \wedge d\Omega_3, \tag{45}$$

where $H_3 = dB + \frac{2\pi}{3n_W}(c(\mathfrak{g}) - c(\mathfrak{h}))\Omega_3$ is the $\mathfrak{u}(1)$ 3-form flux. For the case of interest, $\mathfrak{g} = \mathfrak{su}(2)$, $\mathfrak{h} = 0$ and hence $c(\mathfrak{g}) - c(\mathfrak{h}) = 6$ and $n_W = 2$.

## 4.2 Two-dimensional Effective Action

The effective two-dimensional theory on $T^2$ is obtained by compactifying the six-dimensional theory on a manifold of the form $M = X \times T^2$ for small $X$. The field content in two dimensions is obtained by standard twisted Kaluza-Klein reduction of a single (2,0) $\mathfrak{u}(1)$ tensor multiplet on $X$, by counting harmonic sections of certain bundles. Compactification on a general four-manifold is described, for example, in [41] in Table 1. The choice of the topological twist on the four-manifold $X$ that we are using leads to supersymmetry in the *right-moving*, or, equivalently, *anti-holomorphic*, sector of the effective 2d theory. This is correlated with the convention that the contribution from instantons, which preserve supersymmetry in the 4d theory, has *holomorphic* dependence on the modulus $\tau$ of the $T^2$.

The charge lattice in [41] is the standard charge lattice $\mathbb{Z}$ of a $U(1)$ gauge theory corresponding to the tensor multiplet on the worldvolume of a single M5-brane. In our application, the tensor multiplet really describes the relative motion of a pair of M5-branes. As a result, the normalization of the charge lattice is modified, as we discuss below and in Section 5.2.

We review how the various sigma model fields arise from the KK reduction. For simplicity, consider a simply-connected four-manifold $X$. The 6d scalar fields that transform in the vector representation of $Spin(5)_R$ in the untwisted theory give rise in the twisted theory to $b_2^+$ real non-chiral non-compact 2d scalars $\sigma^i$ plus a single complex non-chiral non-compact scalar $\phi_0$. The 6d fermions transforming in the $(\mathbf{4},\mathbf{4})$ representation of $Spin(5)_R \times Spin(6)$ give rise to $b_2^+$ right-moving Weyl fermions $\psi_+^i$ and a single right-moving Weyl fermion $\chi_+$. The topological twist preserves $\mathcal{N} = (0,2)$ supersymmetry in two dimensions for a generic $X$; this is enhanced to $\mathcal{N} = (0,4)$ for a Kähler $X$.

The 2-form gauge field $B$ with self-dual curvature 3-form gives rise to $b_2 = b_2^+ + b_2^-$ compact chiral real bosons $X_{L,R}^i$ valued in $H^2(X,\mathbb{R})/H^2(X,2\pi\mathbb{Z})$. The quotient arises from the large gauge transformations that change the holonomy of the 2-form gauge field along 2-cycles $C_i \subset X$ by an integer: $\int_{C_i} B \to \int_{C_i} B + 2\pi n_i$ , $n_i \in \mathbb{Z}$. The bosons valued in the subspace of self-dual harmonic forms, $H^{2+}(X) \subset H^2(X,\mathbb{R})$, are right-moving and the bosons valued in $H^{2-}(X)$ are left-moving. In other words, the 2-form field $B$ gives rise to a Narain-like lattice CFT with the lattice $\Gamma = H_2(X,\mathbb{Z})$ being the second homology lattice equipped with the standard geometric intersection form. For a closed four-manifold, $H^2(X,\mathbb{Z}) \cong H_2(X,\mathbb{Z})$ by Poincaré duality and the lattice is self-dual.

When the tensor multiplet describes the tensor branch of 6d type $A_1$ theory, the lattice should be rescaled by $\sqrt{2}$ compared to the case of a single 5-brane considered above because $\Gamma_{SU(2)} = \sqrt{2}\Gamma_{U(1)}$. One way to see this is that the root lattice of $SU(2)$ is $\Lambda = \sqrt{2}\mathbb{Z}$, instead of just $\mathbb{Z}$ when embedded in $\mathbb{R}$ with the standard metric, while its weight lattice is the dual $\Lambda^* = \frac{1}{\sqrt{2}}\mathbb{Z}$. For a general 6d theory on a tensor branch, with string charges living in the lattice[10] $\Lambda^*$, the 2-form fields give rise to 2d lattice CFT for the lattice $\Gamma = \Lambda \otimes H_2(X,\mathbb{Z})$. Note that when $\Lambda$ is not self-dual, $\Gamma$ is not self-dual even for a closed four-manifold. This implies that the effective 2d theory is not *absolute* but *relative,* meaning that instead of a single partition function it has a vector of partition functions labelled by the elements of the coset [42–45]

$$\Gamma/\Gamma^* = \Lambda/\Lambda^* \otimes H^2(X,\mathbb{Z}). \tag{46}$$

This is in agreement with the fact that the 6d theory itself is also in general relative with its relativeness measured by the *defect group* $\Lambda/\Lambda^*$.

Consider now in more detail the case of an $A_1$ theory compactified on $\mathbb{CP}^2$. We have $b_2^- = 0$ and $b_2^+ = 1$. The second homology is generated by the class of $\mathbb{CP}^1 \subset \mathbb{CP}^2$, which has self-intersection number $+1$. The self-dual lattice $H_2(\mathbb{CP}^2,\mathbb{Z})$ can then be identified with the standard lattice $\mathbb{Z}$ and therefore $\Gamma = \sqrt{2}\mathbb{Z}$. The 6d 2-form gauge field gives rise to a single right-moving real compact boson $X_R$ valued in $\mathbb{R}/2\pi\sqrt{2}\mathbb{Z}$, that is, on a circle of radius $\sqrt{2}$.

The compact boson valued in a circle of radius $\sqrt{2}$ can be equivalently thought of as a $\widehat{\mathfrak{u}(1)}_2$ chiral WZW theory, which in turn can be conformally embedded into $\widehat{\mathfrak{su}(2)}_1$. This is another way to understand the rescaling of the lattice by $\sqrt{2}$. Both affine Lie algebras have two integrable modules. Equivalently, both WZW models have two conformal blocks on a

---

[10]Because of the self-duality of the 2-form gauge field, the magnetic string charges are identified with the electric charges and the intersection pairing is the generalization of the Dirac pairing.

torus with characters given by the corresponding lattice theta-functions:

$$
\begin{aligned}
\bar{\chi}_0^{\widehat{\mathfrak{su}(2)}_1}(\tau) &= \bar{\chi}_0^{\widehat{u(1)}_2}(\tau) = \frac{1}{\eta(\tau)} \sum_{n \in \mathbb{Z}} \bar{q}^{n^2}, \\
\bar{\chi}_1^{\widehat{\mathfrak{su}(2)}_1}(\tau) &= \bar{\chi}_1^{\widehat{u(1)}_2}(\tau) = \frac{1}{\eta(\tau)} \sum_{n \in \mathbb{Z}} \bar{q}^{(n+1/2)^2},
\end{aligned}
\tag{47}
$$

where one can already recognize the contributions of abelian anti-instantons which appeared in the holomorphic anomaly equation (11). As explained above, the choice of the module corresponds to the choice of the discrete magnetic flux on $\mathbb{CP}^1 \subset \mathbb{CP}^2$. Note that the lattice point $\frac{m}{\sqrt{2}}$ with $m \in \mathbb{Z}$ corresponds to a vertex operator of the form $e^{\frac{m}{\sqrt{2}} i X_R}$. From the six-dimensional point of view, this is a local two-dimensional operator obtained by wrapping a codimension four defect in the 6d theory on $\mathbb{CP}^1 \subset \mathbb{CP}^2$.

From the general analysis above, we also have three non-compact non-chiral real bosons and two right-moving Weyl fermions. Note that if the compact boson were non-chiral the 2d fields would form the fields of the standard (0,4) sigma model with the target space $\mathcal{X} = (S^1 \times \mathbb{R}^3)/\mathbb{Z}_2 = (\mathbb{C}^* \times \mathbb{C})/\mathbb{Z}_2$ where $\mathbb{Z}_2$ is the $SU(2)$ Weyl symmetry that flips all four directions. There are no left-moving fermions.

The additional data of the sigma-model is the Kalb-Ramond 2-form field[11] $b$ on the target space (not to be confused with the self-dual 2-form field $B$ in 6d). The $b$-field determines the 2d Wess-Zumino term depending on the bosonic fields. This term can be obtained by the reduction of the 6d Skyrmionic string coupling term (43) as follows.

The topological twist on a Kähler $X$ is realized by turning on a non-trivial background for the subgroup of R-symmetry $U(1)_R \equiv Spin(2)_R \subset Spin(5)_R$. The background corresponds to identification of $U(1)_R$ with the diagonal of $U(2)$ holonomy group of $X$. One can assume that the $Spin(2)_R$ rotates the directions 4-5 so that $F^{a_1 a_2} = 0$ unless $a_1, a_2$ are permutation of $4, 5$ and $[F^{45}/(2\pi)] = -[F^{54}/(2\pi)] = c_1(KX) = -c_1(X)$. The non-trivial contribution to the action of the effective 2d theory is given only by the second term in (43). Namely, let $M = \Sigma^2 \times \mathbb{CP}^2$, where $\Sigma^2$ is the 2d spacetime. The KK reduction of 6d bosonic fields to massless 2d fields is explicitly realized as follows. For the $B$-field, we have

$$
B = X_R \, \omega / \sqrt{2}, \tag{48}
$$

where $\omega$ is the Kähler 2-form (which is harmonic and self-dual) normalized such that $\int_{\mathbb{CP}^1} \omega = 1$, and $X_R$ is the right-moving compact boson of the effective 2d theory with the identification $X_R \sim X_R + 2\pi\sqrt{2}$. For the scalar fields, we have:

$$
\begin{aligned}
\Phi^a &= 0, \quad a = 4, 5, \\
\Phi^a &= \phi^a, \quad a = 1, 2, 3,
\end{aligned}
\tag{49}
$$

where the first equation follows from the fact that the canonical bundle of $\mathbb{CP}^2$ has no harmonic sections, and $\phi^a$ are the 2d non-chiral scalar fields valued in $\mathbb{R}^3$.

Performing the partial integration in (43) over the $\mathbb{CP}^2$ yields the bosonic WZ term in 2d:

$$
\begin{aligned}
S_{\text{2d WZ}} &= -i \left( \int_{\mathbb{CP}^2} \frac{F^{45}}{2\pi} \wedge \omega \right) \int_{\Sigma^2} \frac{X_R}{\sqrt{2}} \frac{1}{8\pi} \epsilon_{abc} \partial_i \hat{\phi}^a \partial_j \hat{\phi}^b \hat{\phi}^c \, dx^i \wedge dx^j \\
&= 3i \int_{\Sigma^2} \frac{X_R}{\sqrt{2}} \hat{\phi}^*(\omega_2),
\end{aligned}
\tag{50}
$$

where $\omega_2$ is the volume form on $S^2$ normalized such that $\int_{S^2} \omega_2 = 1$ with $\hat{\phi}^a := \phi^a/\|\phi\|$ understood as a map $\Sigma^2 \to S^2$. In the second equality in (50) we used the fact that the canonical bundle over $\mathbb{CP}^2$ is $O(-3) \to \mathbb{CP}^2$, so that $[F^{45}/(2\pi)] = c_1(K\mathbb{CP}^2) = -c_1(\mathbb{CP}^2) = -3[\omega]$.

---

[11]To be precise, the 2-form description is only local; globally $b$ should be understood as a gerbe connection.

Let us denote the coordinate on $S^1$ of the target space[12] of the effective theory by $Y^0$ and the coordinates on $\mathbb{R}^3$ as $Y^a$ ($a = 1,2,3$), with norm $\|Y\|$. From (50) we deduce that the Kalb-Ramond field $b$ on target space is given by (the choice of normalization is

$$b = 4\pi \cdot \frac{3}{\sqrt{2}} Y^0 \omega_2 = \frac{3}{2\sqrt{2}} Y^0 \epsilon_{abc} \hat{Y}^a d\hat{Y}^b \wedge d\hat{Y}^c \tag{51}$$

with corresponding 3-form flux

$$h = db = 4\pi \cdot \frac{3}{\sqrt{2}} dY^0 \wedge \omega_2 = \frac{3}{2\sqrt{2}} dY^0 \wedge \epsilon_{abc} \hat{Y}^a d\hat{Y}^b \wedge d\hat{Y}^c , \tag{52}$$

where $\hat{Y}^a := Y^a / \|Y\|$. The choice of normalization is such that $\int h \in 8\pi^2 \mathbb{Z}$ and will be consistent with the normalization of the action in Section 4.3. The compactification of the 6d Wess-Zumino term (40) to 2d is not relevant for our computation of the holomorphic anomaly because it produces the term

$$9 \cdot 2\pi \int_{\Xi^3} \Omega_1 d\Omega_1, \qquad d\Omega_1 = \hat{\phi}^*(\omega_2), \tag{53}$$

where $\partial \Xi^3 = \Sigma^2$. This term is the 2d analog of the 6d Hopf-Wess-Zumino term (40) introduced in [18]. It is well defined since the integral of the form $\Omega_1 d\Omega_1$ over a closed 3-manifold is an integer, the Hopf invariant of the map from this 3-manifold to $S^2$. The term (53) does not contain the compact boson $X_R$. The fermionic partner of this term therefore does not affect the saturation of zero-modes.

To compute the holomorphic anomaly we need not the bosonic WZ term (50) itself, but rather its fermionic partner. The effective two-dimensional theory is a fairly standard heterotic sigma model with target $\mathcal{X} = (S^1 \times \mathbb{R}^3)/\mathbb{Z}_2$, except that the bosonic field valued in $S^1$ is only right-moving. This does not affect supersymmetrization of the action because the supersymmetry generators act only on the right-movers. One can thus use known results about heterotic sigma models reviewed below to deduce the relevant fermionic terms.

## 4.3 Review of (0,1) Nonlinear Sigma Model

The goal of this section is to review basic facts about (0,1) Nonlinear Sigma models and to fix various normalizations that will be relevant for our calculation of the holomorphic anomaly. We can restrict ourselves to the class of theories containing only scalar multiplets $(\phi^i, \psi^i)$ composed of bosonic scalars $\phi^i$ and right-moving Majorana-Weyl spinors $\psi^i$. The scalars play the role of local coordinates in the target space while spinors $\psi^i$ are valued in the tangent bundle. In general the theory can also have left-moving Majorana-Weyl spinors valued in a real vector bundle over the target space; however for our purposes this generalization is not necessary.

Let $g_{ij}$ and $b = \frac{1}{2} b_{ij} dx^i \wedge dx^j$ be the metric and Kalb-Ramond 2-form on the target space. The theory is anomaly-free and, in particular, invariant under symmetries of the target, provided that $w_1(T\mathcal{X}) = w_2(T\mathcal{X}) = 0$ and $p_1(T\mathcal{X})/2 = 0$; here $w_i$ denote Stiefel-Whitney classes and $p_1$ denotes the first integral Pontryagin class. Denote $\partial := \partial/\partial z$, $\bar{\partial} := \partial/\partial \bar{z}$. The action of the theory in Euclidean spacetime with a local complex coordinate $z$ then reads[13] [47, 48]:

$$S_{2d} = \frac{1}{4\pi} \int d^2z \left( (g_{ij}(\phi) + b_{ij}(\phi)) \partial \phi^i \bar{\partial} \phi^j + g_{ij} \psi^i \partial \psi^j - (\Gamma_{ijk} + \tfrac{1}{2} h_{ijk}) \psi^k \psi^i \partial \phi^j \right), \tag{54}$$

---

[12]To avoid confusion, we use different symbols for the coordinates on the target space and the fields on $\Sigma^2$ *valued* in the target space.

[13]Our normalization corresponds to the choice $\alpha' = 2$ in the string theory literature [46].

where $d^2z := idz d\bar{z}$, $\Gamma_{ijk}$ are Christoffel symbols of the Levi-Cevita connection, and

$$h = db = \frac{1}{3!} h_{ijk} dx^i \wedge dx^j \wedge dx^k, \qquad h_{ijk} = \partial_i b_{jk} + \partial_j b_{ki} + \partial_k b_{ij} \tag{55}$$

is the 3-form flux of $b$. The term containing the scalar fields and the Kalb-Ramond field in the target is the Wess-Zumino term, which can be recast as

$$S_{2d\,WZ} = \frac{i}{4\pi} \int_{\Sigma^2} \phi^*(b) = \frac{i}{4\pi} \int_{\Xi^3} \phi^*(h), \tag{56}$$

where $\partial \Xi^3 = \Sigma^2$. We normalize the supercharge so that it acts on the fields as follows:

$$\begin{aligned}
[Q, \phi^i] &= \psi^i, \\
\{Q, \psi^i\} &= -\bar{\partial} \phi^i.
\end{aligned} \tag{57}$$

The right-moving energy momentum-tensor and the supercurrent are given by (cf. [49])

$$\begin{aligned}
\bar{T} &= -\tfrac{1}{2} g_{ij} \bar{\partial} \phi^i \bar{\partial} \phi^j - \tfrac{1}{2} g_{ij} \psi^i \bar{\partial} \psi^j + \tfrac{1}{2} \partial_k g_{ij} \psi^k \psi^j \bar{\partial} \phi^i + \tfrac{1}{4} h_{ijk} \psi^i \psi^j \bar{\partial} \phi^k, \\
\bar{G} &= i \left( g_{ij} \psi^i \bar{\partial} \phi^j - \tfrac{1}{3!} h_{ijk} \psi^i \psi^j \psi^k \right).
\end{aligned} \tag{58}$$

The normalization of the energy-momentum tensor $\bar{T}$ is fixed by the definition $\bar{T} = 2\pi \delta S_{2d} / \delta h_{\bar{z}\bar{z}}$, where $h_{\bar{z}\bar{z}}$ is the corresponding component of the metric on the 2d space-time. The normalization of the supercurrent $\bar{G}$ is then fixed by the relation

$$\{Q, \bar{G}\} = 2i \, \bar{T} \tag{59}$$

with the supercharge defined above. These formulae compactly summarize the supersymmetrization in two dimensions, which is much simpler than the supersymmetrization in four-dimensions described in Appendix A. In particular, there is no need for auxiliary fields.

The normalization of the path integral for the theory on a torus can now be fixed by requiring that it coincides with the corresponding trace over the Hilbert space on the circle:

$$Z(\tau) = \mathrm{Tr}(-1)^F q^{L_0} \bar{q}^{\bar{L}_0}, \tag{60}$$

where we consider periodic-periodic boundary conditions on the fermions, that is, odd spin structure. Consider in particular the theory of $D$ free $(0,1)$ scalar multiplets described by the metric $g_{ij} = \delta_{ij}$ and vanishing $b_{ij}$. The theory then factorizes into the theory of $D$ free bosons and $D$ free Majorana-Weyl fermions. With the choice of normalization of the action above, the bosons and fermions satisfy the following operator product expansions:

$$\phi^i(z) \phi^j(0) \sim -\delta^{ij} \log|z|^2, \tag{61}$$

$$\psi^i(z) \psi^j(0) \sim \frac{\delta^{ij}}{\bar{z}}. \tag{62}$$

The partition function of the bosons on the torus reads

$$Z_{bos}(\tau) = \frac{V_D}{(8\pi^2 \tau_2)^{D/2}} |\eta(\tau)|^{-2D}, \tag{63}$$

where $V_D$ is the regularized volume of the target space. The prefactor originates from the trace over the momentum space [46] and is given by

$$V_D \int \frac{d^D k}{(2\pi)^D} e^{-2\pi \tau_2 k^2} = \frac{V_D}{(8\pi^2 \tau_2)^{D/2}}. \tag{64}$$

The partition function of free fermions is zero due to the presence of zero-modes. To get a nonzero answer one can consider instead a one-point function of the product of all spinor fields:

$$\langle : \psi^1 \psi^2 \ldots \psi^D : \rangle_{\text{fer}} = \text{Tr}(-1)^F \psi_0^1 \psi_0^2 \ldots \psi_0^D \, q^{L_0} \bar{q}^{\bar{L}_0} \,, \tag{65}$$

where $\psi_0^i$ are zeroth components of the fermionic fields $\psi^i(z) = \sum_n \psi_n^i z^{-n-1/2}$ in the Ramond sector. They satisfy the following anti-commutation relation:

$$\{\psi_0^i, \psi_0^j\} = \delta^{ij}. \tag{66}$$

The ground states in the Hilbert space on the circle therefore form the standard spinor representation of the Clifford algebra with identification $\psi^i = 2^{-1/2}\gamma^i$, where $\gamma^i$ are the standard Eulidean gamma-matrices satisfying $\{\gamma^i, \gamma^j\} = 2\delta^{ij}$. When $D$ is even the fermion parity is non-anomalous and on the ground states can be represented by

$$(-1)^F = \gamma_5 = (-i)^{D/2}\gamma^1\gamma^2\ldots\gamma^D. \tag{67}$$

Therefore the trace over the ground states reads

$$\text{Tr}_{\text{ground}}(-1)^F \psi_0^1 \psi_0^2 \ldots \psi_0^D = i^{D/2} 2^{-D/2} \, \text{Tr}_{\text{ground}} \gamma_5^2 = i^{D/2}. \tag{68}$$

The full Hilbert space is obtained by action of the creation operators $\psi_{-n}^i$, $n > 0$ on the ground states. Therefore, the one-point function on the torus reads

$$\langle : \psi^1 \psi^2 \ldots \psi^D : \rangle_{\text{fer}} = i^{D/2} \overline{\eta(\tau)}^D. \tag{69}$$

Even though this expression is obatined for even $D$, it can be used to define the normalization of the torus one-point function for arbitrary $D$. Note that for odd $D$ the sign of $i^{D/2}$ is ambiguous which corresponds to the mod 2 fermion parity anomaly. When an even number of fermions is separated into two groups each containing an odd number of fermions, it is necessary to choose the signs consistently. To be concrete, we choose the convention $i^{D/2} = e^{\pi i D/4}$.

Consider now a more general $(0,1)$ sigma-model, but in a limit when the target space $\mathcal{X}$ has large radius of curvature. There is a standard one-to-one correspondence between local observables of the form $\mathcal{O} =: f_{i_1\ldots i_k}(\phi)\psi^{i_1}\ldots\psi^{i_k} :$ and $k$-forms on the target space $f = f_{i_1\ldots i_k}(\phi)d\phi^{i_1}\ldots d\phi^{i_k} \in \Omega^k(\mathcal{X})$. Using this correpondence and the above results on the free fields, we obtain the following formula for the one-point function in the large radius limit:

$$\langle : f_{i_1\ldots i_k}(\phi)\psi^{i_1}\ldots\psi^{i_k} : \rangle = \frac{i^{D/2}}{(8\pi^2\tau_2)^{D/2}\,\eta(\tau)^D} \int_{\mathcal{X}} f. \tag{70}$$

This fixes the normalization of the measure for the zero-modes in the path-integral which will be used in the next section. Finally, note that the supercharge $Q$ in (57), when restricted to the zero-modes, acts as the exterior derivative on $\Omega^*(\mathcal{X})$ under this correspondence.

## 4.4 Holomorphic Anomaly from the Sigma Model

The elliptic genus of a compact target $\mathcal{X}$ is given by the partition function of a $(0,1)$ supersymmetric sigma model with periodic boundary conditions for fermions in both directions on $T^2$. When $\mathcal{X}$ is not compact, the partition function is in general not holomorphic and has a holomorphic anomaly. If $\mathcal{X}$ has a "boundary"[14] $\mathcal{Y}$, then the holomorphic anomaly is governed

---

[14]We use the term "boundary" informally. For example, if $\mathcal{X}$ is asymptotic to $\mathbb{R}^N$ at infinity for some $N$, then by $\mathcal{Y}$ we mean a large sphere $S^{N-1}$ near infinity in $\mathcal{X}$.

by the one-point function of the supercharge in a sigma model with target $\mathcal{Y}$ [14]. More precisely, let $Z_{\mathcal{X}}$ be the partition function of a heterotic sigma model with target $\mathcal{X}$ on a torus with complex structure $\tau$. Then[15]

$$\frac{\partial Z_{\mathcal{X}}}{\partial \bar{\tau}} = \langle -2\pi i \bar{T}(z_0)\rangle_{\mathcal{X}} = \langle -\pi\{Q, \bar{G}(z_0)\}\rangle_{\mathcal{X}} = \frac{-e^{\pi i/4}}{\sqrt{8\,\tau_2}\eta(\tau)}\langle \bar{G}(z_0)\rangle_{\mathcal{Y}}, \tag{71}$$

where $\langle \mathcal{O}(z_0)\rangle$ is a path integral with an insertion of operator $\mathcal{O}(z_0)$ at an arbitrary point $z_0$.

This formula can be understood as follows. The first equality follows from the trace (60) or from the definition of $\bar{T}$ and the fact that the change of a complex structure can be related to the change of metric. The second equality in (71) uses the relation $2i\,\bar{T} = \{Q, \bar{G}\}$ between the energy momentum tensor $\bar{T}$ and supercurrent $\bar{G}$ via the supersymmetry transformation $Q$. The last equality in (71) is more subtle and can be argued as follows.

The partition function $Z_{\mathcal{X}}$ is obtained by integrating over the space of all maps $\phi: T^2 \to \mathcal{X}$ along with fermion fields $\psi \in \Gamma(S_+(T^2) \otimes \phi^* T\mathcal{X})$. The path integral has zero-modes consisting of a map $\phi: T^2 \to \mathcal{X}$ together with a constant $\psi$ field. We write $(\phi_0, \psi_0)$ for such a constant pair. Supersymmetry localizes the path integral on the space of constant pairs. This means that the path integral can be evaluated by integrating out other modes to construct a measure $f(\phi_0, \psi_0)$, which must then be integrated over $\phi_0, \psi_0$. In an ordinary sigma model, to compute $f$, it suffices to integrate out the other modes in a 1-loop approximation because $Z_{\mathcal{X}}$ does not depend on the metric of $\mathcal{X}$. Hence, one can scale up the metric of $\mathcal{X}$ so that the 1-loop computation over nonzero modes is exact. In the present situation, we have to modify this procedure slightly because there is a right-moving chiral boson with no tunable parameter such as a variable metric. However, the effects of this mode can be treated exactly, treating other nonzero modes in the one-loop approximation.

Now we come to the main point of the argument. We have already explained in section 4.3 that an operator of the form $\mathcal{O}_f =: f_{i_1\dots i_k}(\phi)\psi^{i_1}\dots\psi^{i_k}:$ corresponds to a differential form $f = f_{i_1\dots i_k}(\phi)d\phi^{i_1}\dots d\phi^{i_k} \in \Omega^k(\mathcal{X})$, and that the path integral $\langle \mathcal{O}_f\rangle$ is the integral $\int_X f$ of the differential form $f$, times some additional factors explained in eqn. (70). Moreover, acting on operators that are related to differential forms in this way, $Q$ corresonds to the exterior derivative $d$. To claim that a $Q$-exact term does not contribute to the path integral amounts to claiming that $\int_X f$ is invariant under $f \to f + dg$. But in general there is the possibility of a surface term, because by Stokes's theorem $\int_{\mathcal{X}} dg(\phi_0, \psi_0) = \int_{\mathcal{Y}} g(\phi_0, \psi_0)$, where $\mathcal{Y} = \partial\mathcal{X}$. The anomaly captures the failure of the naive argument of vanishing of $Q$-exact terms, and hence reduces to an integral over $\mathcal{Y}$.

Since we are interested in the expectation value of $\{Q, \bar{G}\}$, we have $f = dg$ where $g$ is a function of $\phi_0, \psi_0$ that is computed by integrating out all modes of a chiral boson and nonzero modes for all other fields in the path integral for $\langle \bar{G}\rangle$. We are only interested in evaluating $g$ on $\mathcal{Y}$. Most of the modes that appear in the evaluation of $g(\phi_0, \psi_0)$ are the modes that would appear in evaluating $\langle \bar{G}\rangle$ in a sigma-model with target $\mathcal{Y}$, with one important exception. In the sigma-model with target $\mathcal{X}$, there is a bosonic field $\phi_\perp$ that describes the motion normal to $\mathcal{Y}$, and a corresponding fermion partner $\psi_\perp$. These modes are absent in the sigma-model with target $\mathcal{Y}$, so we have to include them separately. The partition function of the left-moving modes of $\phi_\perp$ is the factor $1/\eta(\tau)$ in eqn. (71). The right-moving modes of $\phi_\perp$ cancel the nonzero modes of $\psi_\perp$. The zero-modes of $\phi_\perp$ and $\psi_\perp$ are eliminated when we use Stoke's theorem and replace $\int_{\mathcal{X}} dg$ with $\int_{\mathcal{Y}} g$, except for a normalization factor

$$\frac{e^{\pi i/4}}{\sqrt{8\pi^2\tau_2}} \tag{72}$$

---

[15]If the sigma model with target $\mathcal{X}$ has $(0, k)$ supersymmetry with $k > 1$, then in the following formula, for $Q$, one can use any supersymmetry for which the following derivation applies.

explained in Section 4.3. Altogether, one obtains precisely the right hand side of (71).

In our case, $\mathcal{X} = (\mathbb{R}^3 \times S^1)/\mathbb{Z}_2$ and $\mathcal{Y} = (S^2_R \times S^1)/\mathbb{Z}_2$, where $S^2_R$ is a sphere of very large radius $R$.[16] In the limit $R \to \infty$, the fermions and bosons appearing in a sigma model with target $\mathcal{Y}$ can be treated as being free. We thus obtain

$$
\begin{aligned}
\langle G_+ \rangle_{\mathcal{Y}} &= \left\langle -i\frac{1}{3!}h_{ijk}\psi^i\psi^j\psi^k + ig_{is}\partial_+\phi^i\psi^s \frac{1}{4\pi}\int d^2z\left(\left(\Gamma_{ijk} + \frac{h_{ijk}}{2}\right)\psi^k\psi^j\partial_-\phi^j\right)\right\rangle_{\mathcal{Y}} \\
&= -\frac{i}{3!}\langle h_{ijk}\psi^i\psi^j\psi^k\rangle_{\mathcal{Y}},
\end{aligned}
\tag{73}
$$

where we used the fact that

$$
\int d^2x \langle \partial_+\phi^i(x)\partial_-\phi^j(x')\rangle = 0
\tag{74}
$$

in free field theory. When we interpret the fermion zero-modes as 1-forms on $\mathcal{Y}$, the coupling $h_{ijk}\psi^i\psi^j\psi^k$ just becomes a 3-form $h = \frac{1}{3!}h_{ijk}d\phi^i d\phi^j d\phi^k$ that has to be integrated over $\mathcal{Y}$. Therefore, the result will only depend on the cohomology class of the Wess-Zumino coupling. Using the expression (52) for $h$, and taking into account the periodicity $Y^0 \sim Y^0 + 2\pi\sqrt{2}$, we obtain

$$
\int_{\mathcal{Y}} h = \frac{1}{2}\cdot 4\pi \cdot 3 \int_{S^2}\omega_2 \int_{S^1}\frac{dY^0}{\sqrt{2}} = 12\pi^2.
\tag{75}
$$

Here we only need $\int_{S^2}\omega_2 = 1$ and not the explicit expression for the 2-form $\omega_2$.

The nonzero modes of the fermion contribute $\overline{\eta(\tau)^3}$. It is also necessary to include the normalization phase $e^{3\pi i/4}$ for the fermionic zero-modes explained in Section 4.3. The nonzero modes of the bosons valued in $S^2$ contribute $1/(8\pi^2\tau_2|\eta(\tau)|^4)$ with normalization also explained in Section 4.3. The nonzero modes of the chiral boson valued in $S^1$ contribute $\bar{\chi}^{\widehat{\mathfrak{u}(1)_2}}_v(\tau)/(2\pi\sqrt{2})$, which differs from its partition function (47) by factoring out the integral over the zero-mode $\int dY^0 = 2\pi\sqrt{2}$. Here $v \in \mathbb{Z}_2$ corresponds to the discrete flux of the $SO(3)$ gauge field on $\mathbb{CP}^1 \subset \mathbb{CP}^2$. Combining all the contributions we obtain

$$
\begin{aligned}
\langle \bar{G} \rangle_{\mathcal{Y}} &= -i\left(\int_{\mathcal{Y}}h\right)\cdot e^{3\pi i/4}\overline{\eta(\tau)^3}\cdot \frac{1}{8\pi^2\tau_2\eta(\tau)^2\overline{\eta(\tau)^2}}\cdot \frac{\bar{\chi}^{\widehat{\mathfrak{u}(1)_2}}_v(\tau)}{2\pi\sqrt{2}} \\
&= \frac{-3i\cdot e^{3\pi i/4}}{4\pi\sqrt{2}\tau_2\eta(\tau)^2}\cdot \sum_{n\in\mathbb{Z}}\bar{q}^{(n+v/2)^2}.
\end{aligned}
\tag{76}
$$

The anti-holomorphic eta-functions cancel between bosons and fermions due to right-moving supersymmetry. The anti-holomorphic theta function is the contribution of the "winding modes" of the compact chiral boson. Note that, as in 4d calculation, the overall sign in (76) is somewhat ambiguous and depends on the choice of orientation in the space of zero-modes. We do not address the question of fixing it in this paper. Nevertheless, combining (76) with (71) we obtain

$$
\frac{\partial Z_v}{\partial\bar{\tau}} = \frac{3}{16\pi i\tau_2^{3/2}\eta(\tau)^3}\sum_{n\in\mathbb{Z}}\bar{q}^{(n+v/2)^2},
\tag{77}
$$

which is in agreement with the expected result (10)-(11).

---

[16]Near the origin of $\mathcal{X} = (\mathbb{R}^3 \times S^1)/\mathbb{Z}_2$, a good description is likely to involve additional degrees of freedom. However, since this region is compact, it does not contribute to the anomaly

# 5 Generalizations

We now turn to possible generalizations. In Section 5.1 we consider a general Kähler manifold with $b_2^+ = 1$, $b_1 = 0$ and compute the holomorphic anomaly of the Vafa-Witten partition function. In Section 5.2 we consider $SU(N)$ gauge theory realized on multiple M5-branes. The holomorphic anomaly can again be traced to the Wess-Zumino term in the effective action on the Coulomb branch where $SU(N)$ is spontaneously broken to $SU(N_1) \times SU(N_2) \times U(1)$ with $N = N_1 + N_2$. Finally, in Section 5.3 we briefly discuss other twists.

## 5.1 Vafa-Witten Theory on Kähler Manifolds with $b_2^+ = 1$, $b_1 = 0$

Consider $SO(3)$ gauge theory on a general Kähler manifold $X$ with $b_2^+ = 1$ and $b_1 = 0$ obtained by wrapping two M5-branes on $X$ in M-theory on $KX \times T^2 \times \mathbb{R}^3$, where, as before, $KX$ is the total space of the canonical bundle over $X$. Unlike in the case $X = \mathbb{CP}^2$, we will in general have a nonzero $b_2^-$ with interesting modifications.

Consider first the two-dimensional point of view. The field content of the effective (0,4) sigma model can be obtained by Kaluza-Klein reduction of the 6d (2,0) tensor multiplet as in Section 4.2 following [41]. The only additional fields are $b_2^-$ left-moving compact bosons because the relation (48) is replaced by the following decomposition of the six-dimensional 2-form field:

$$B = \frac{1}{\sqrt{2}} \sum_{i=1}^{b_2^-} X_L^i h_i^- + \frac{1}{\sqrt{2}} X_R \omega, \tag{78}$$

where $h_i^-$ are the generators of $H^{2-}(X, \mathbb{R})$ and $\omega$ is the Kähler 2-form, normalized such that

$$\int_X \omega \wedge \omega = 1, \qquad \int_X h_i^- \wedge h_j^- = -\delta_{ij}, \qquad \int_X h_i^- \wedge \omega = 0. \tag{79}$$

The fields $X_L^i$ and $X_R$ are components of left-moving and right-moving compact bosons; $(X_L, X_R)$ is valued in the torus $H^{2-}(X, \mathbb{R}) \oplus H^{2+}(X, \mathbb{R})/H^2(X, \mathbb{Z})$.

Thus, the compact bosons of the 2d theory form Narain lattice CFT associated to the indefinite lattice $\Lambda \otimes H^2(X, \mathbb{Z})$. As before, $\Lambda \cong \sqrt{2}\mathbb{Z}$ denotes the root lattice of $SU(2)$. What is important is that the intersection form still has a single negative eignvalue. When $b_2^- > 0$ the theory has non-trivial moduli depending on the conformal class of the metric of $X$. It has special "walls" that appear when the intersection $H^{2,+}(X, \mathbb{R}) \cap (H^2(X, \mathbb{Z}) \otimes \mathbb{R})$ has nonzero elements, corresponding to abelian instantons.

The fermionic and non-compact bosonic fields will be the same as in the case $X = \mathbb{CP}^2$. Let us also assume that, as in the $X = \mathbb{CP}^2$ case, the canonical class lies inside $H^{2,+}(X, \mathbb{R})$. That is $c_1(X) \cdot [h_i^-] = 0$, $\forall i$, where $\cdot$ denotes the intersection pairing $H^2(X, \mathbb{R}) \otimes H^2(X, \mathbb{R}) \to \mathbb{R}$. The analysis of the holomorphic anomaly is then analogous to the $\mathbb{CP}^2$ case and the final result is modified to

$$\frac{\partial Z_\nu}{\partial \bar\tau} = \frac{(c_1(X) \cdot [\omega])}{16\pi i \, \tau_2^{3/2} \eta(\tau)^{3+b_2^-}} \sum_{\substack{n \in \frac{1}{2}H^2(X,\mathbb{Z}) \\ n = \frac{\nu}{2} \mod H^2(X,\mathbb{Z})}} \bar{q}^{n^+ \cdot n^+} q^{-n^- \cdot n^-}, \tag{80}$$

where

$$n^+ := (n \cdot [\omega])[\omega], \qquad n^- := \sum_{i=1}^{b_2^-} (n \cdot [h_i^-])[h_i^-], \tag{81}$$

and

$$\nu \in H^2(X, \mathbb{Z}_2) \tag{82}$$

is the $\mathbb{Z}_2$ magnetic flux of the $SU(2)/\mathbb{Z}_2 \cong SO(3)$ gauge field. This is in agreement with the prediction for the holomorphic anomaly [50, 51] obtained from a different perspective.

There are two modifications in (80) compared to the $\mathbb{CP}^2$ case (71). The first is the overall factor, $(c_1(X)\cdot[\omega])$ which is the straightforward generalization of the factor that appears after reduction of 6d Skyrmionic string term to 2d WZ term as in (50). The second is the contribution

$$\frac{q^{-n^-\cdot n^-}}{\eta(\tau)^{b_2^-}} \tag{83}$$

from left-moving compact bosons $X_L^i$.

Consider for example the case $X = \mathbb{CP}^1 \times \mathbb{CP}^1$ with $b_2^+ = b_2^- = 1$. The $H^2(X, \mathbb{Z})$ lattice has two generators $e_1$ and $e_2$, Poincaré dual to 2-cycles pt $\times \mathbb{CP}^1$ and $\mathbb{CP}^1 \times$ pt. Their intersection numbers are:

$$e_1 \cdot e_1 = e_2 \cdot e_2 = 0, \qquad e_1 \cdot e_2 = 1. \tag{84}$$

In terms of this basis, the first Cherm class of the tangent bundle reads

$$c_1(X) = 2e_1 + 2e_2. \tag{85}$$

The orthonormal basis in $H^{2,+}(X, \mathbb{R}) \oplus H^{2,-}(X, \mathbb{R})$ is given by

$$\omega = \frac{e_1}{R} + \frac{R e_2}{2}, \qquad h^- = \frac{e_1}{R} - \frac{R e_2}{2}, \tag{86}$$

where $R^2/2$ is the ratio of the areas of two $\mathbb{CP}^1$'s. In this case the effective two-dimensional theory is closely related to a $(0,1)$ sigma model with $S^1 \times \mathbb{R}^3$ target, where $R$ is the radius of $S^1$. The difference comes from rescaling of the lattice of winding numbers and momenta along $S^1$ by the overall $\sqrt{2}$ factor. The condition $c_1(X)\cdot[h^-] = 0$ is satisfied if $R = \sqrt{2}$. The formula (80) then reads

$$\frac{\partial Z_\nu}{\partial \bar{\tau}} = \frac{1}{4\sqrt{2}\pi i \tau_2^{3/2} \eta(\tau)^4} \sum_{\substack{n \in \mathbb{Z}^2 \\ n = \nu \mod 2}} \bar{q}^{(n_1+n_2)^2/8} q^{(n_1-n_2)^2/8}, \tag{87}$$

where $\nu \in \mathbb{Z}_2^2$.

The analysis in the gauge theory region is similar. The overall factor of the 3-form integrated over the boundary $S^3$ in the space of bosonic zero-modes changes from 3 to $(c_1(X)\cdot[\omega])$. The contribution of the abelian and point-like instantons in (32) is replaced with

$$\frac{1}{\eta(\tau)^{\chi(X)}} \sum_{\substack{n \in \frac{1}{2}H^2(X,\mathbb{Z}) \\ n = \frac{1}{2}\nu \mod H^2(X,\mathbb{Z})}} \bar{q}^{n^+\cdot n^+} q^{-n^-\cdot n^-}, \tag{88}$$

where $\chi(X) = 3 + b_2^-$. This yields the same result (80) obtained in the sigma model region.

If $c_1(X)\cdot[h_i^-] \neq 0$ the analysis both in four and two dimensions will be slightly modified. In the gauge theory region the supersymmetrization of the Wess-Zumino term (performed in Appendix A for the case of $\mathbb{CP}^2$) will lead to an extra term in the action of the form

$$\int_X \frac{F_R \wedge F_A^- H}{2\pi |\Phi|^2}, \tag{89}$$

where $F_R$ is the background R-symmetry curvature, which satisfies $[F_R/(2\pi)] = -c_1(X)$, $F_A^-$ is the anti-selfdual part of the gauge field strength, and $H$ is the scalar auxiliary field. This in turn will lead to an extra term proportional to $\mathcal{H}(c_1(X)\cdot n^-)$ in (30). The result of the

finite-dimensional Gaussian integration in (35) will then be modified by an extra term of the form

$$\tau_2^{-1/2}(c_1(X) \cdot n^-)(n \cdot [\omega]) \bar{q}^{\,n^+ \cdot n^+} q^{-n^- \cdot n^-}. \tag{90}$$

Such an extra term in holomorphic anomaly was already observed in [52] from a different approach[17]. Similarly, in the sigma-models region, when $c_1(X) \cdot [h_i^-] \neq 0$, the 3-form $\phi^*(h)$ will have an extra term proportional to $\sum_i (c_1(X) \cdot [h_i^-]) dX_L^i \hat{\phi}^*(\omega_2)$. The contribution (90) will then arise from the second term in (73) which will be non-vanising, as $\langle \partial X_R \bar{\partial} X_L^i \rangle \neq 0$ for compact bosons $X_R$ and $X_L^i$.

## 5.2 Holomorphic Anomaly for $SU(N)/\mathbb{Z}_N$ Gauge Theory

Another generalization is to consider Vafa-Witten theory for the gauge group $SU(N)/\mathbb{Z}_N$. This theory can be obtained by compactifying on $T^2$ the six-dimensional (2,0) model of type $A_{N-1}$, which can be realized in M-theory by a stack of $N$ M5-branes with the center of mass degrees of freedom removed. For simplicity, consider $X = \mathbb{CP}^2$. By arguments similar to the above, we expect that the contributions to the holomorphic anomaly come from the boundary of the non-compact space of bosonic zero-modes. The latter can originate from topologically twisted scalar bosonic fields serving as coordinates on the Coulomb branch, where the gauge group is spontaneously broken to some subgroup.

Consider breaking $SU(N)$ by a vacuum expectation value of adjoint scalar fields proportional to the traceless matrix

$$T = \frac{1}{\sqrt{N_1 N_2 N}} \left( \begin{array}{c|c} N_2 \mathbf{1}_{N_1 \times N_1} & \mathbf{0}_{N_1 \times N_2} \\ \hline \mathbf{0}_{N_2 \times N_1} & -N_1 \mathbf{1}_{N_2 \times N_2} \end{array} \right), \tag{91}$$

with the standard normalization $\mathrm{Tr}(T^2) = 1$; here $N_1 + N_2 = N$. An $SU(N_1) \times SU(N_2) \times U(1)$ subgroup that commutes with $T$ is left unbroken. Breaking to smaller subgroups can be realized recursively.

The analysis above for $SU(2)/\mathbb{Z}_2$ can be repeated with the scalar fields $\Phi^i$ in the vector multiplet of the unbroken $U(1)$ and their superpartners. The overall coefficients in front of the 6d Skyrmionic string term (43) and 4d WZ term (21) are now given by

$$\frac{n_W}{2} = \frac{(N_1 + N_2)^2 - 1 - (N_1^2 - 1 + N_2^2 - 1 + 1)}{2} = N_1 N_2. \tag{92}$$

This modifies accordingly the overall coefficient on the right hand side of the holomorphic anomaly equation. The integration over the boundary $S^3$ in the space of the bosonic zero-modes of the 4d theory on $\mathbb{CP}^2$ (or boundary $S^2 \times S^1$ in the effective 2d theory), together with summation over all possible partitions $N = N_1 + N_2$ and discrete fluxes of $SU(N_1)/\mathbb{Z}_{N_1} \times SU(N_2)/\mathbb{Z}_{N_2} \times U(1)$ consistent with a given flux of $SU(N)/\mathbb{Z}_N$, gives

$$\frac{\partial Z_v^{SU(N)/\mathbb{Z}_N}(\mathbb{CP}^2)}{\partial \bar{\tau}} = \frac{3}{16\pi i \tau_2^{3/2} \eta(\tau)^3} \times$$

$$\sum_{N_1+N_2=N} N_1 N_2 \sum_{\substack{v_1 \in \mathbb{Z}_{N_1} \\ v_2 \in \mathbb{Z}_{N_2}}} \left( \sum_{\substack{n \in NN_1 N_2 \mathbb{Z} + NN_1 v_2 \\ +NN_2 v_1 + N_1 N_2 v}} \bar{q}^{\frac{n^2}{2NN_1 N_2}} \right) Z_{v_1}^{SU(N_1)/\mathbb{Z}_{N_1}}(\mathbb{CP}^2) Z_{v_2}^{SU(N_2)/\mathbb{Z}_{N_2}}(\mathbb{CP}^2). \tag{93}$$

---

[17]We would like to thank J. Manschot and G. Moore for bringing it to our attention.

The theta function appearing in this formula corresponds to sum over the weight lattice

$$\Lambda^*_{U(1)} = \frac{\mathbb{Z}}{\sqrt{N_1 N_2 N}} \tag{94}$$

of the unbroken $U(1)$ inside the weight lattice $\Lambda^*_{SU(N)}$ of $SU(N)$. The normalization is fixed by the normalization of $T$ in (92) above.

This is in agreement with the general prediction in [50,51,53–55] (where the gauge group is $U(N)$ instead of $SU(N)$). By applying this formula recursively one can conclude that the holomorphic limit of $Z_\nu^{SU(N)/\mathbb{Z}_N}$ is a depth $N-1$ mock modular form.

## 5.3 Other Twists

For completeness, we include the computation of the holomorphic anomaly in the partition function for the B and the C twist. The analysis of the 2d sigma model is very similar for to the one for the A twist and leads to a noncompact theory. The analysis on the Coulomb branch is also similar. However, it turns out that the holomorphic anomaly for the partition function actually vanishes for these twists. It is conceivable that some other observables of the twisted theory have nonvanishing holomorphic anomaly and exhibit mock modularity.

*The C Twist*

This twist is only possible on spin manifolds, because the $\mathcal{N}=4$ theory, considered as a $\mathcal{N}=2$ theory, has matter in the adjoint representation. The corresponding sigma model has $(0,1)$ supersymmetry in two dimensions, which for Kähler manifolds is enhanced to $(0,2)$.

The simplest example is $X = \mathbb{CP}^1 \times \mathbb{CP}^1$. Since its signature is zero, for generic metric there will be no harmonic spinors. The effective theory contains a single non-chiral non-compact boson, and also a left-moving chiral boson $X_L$ and a right-moving chiral boson $X_R$, as in the example at the end of Section 5.1). The non-compact boson and $X_R$ have super-partners: two right-moving Majorana-Weyl fermions $\psi_1, \psi_2$. Unlike in the case of the A twist, the Wess-Zumino term does not play a role as the target space is two-dimensional. The boundary theory $\mathcal{Y}$ contains a single fermionic mode that can be saturated by the supercurrent. However, its expectation value turns out to vanish:

$$\frac{\partial Z_\nu(\mathbb{CP}^1 \times \mathbb{CP}^1)}{\partial \bar{\tau}} \propto \langle \bar{G} \rangle_\mathcal{Y} \propto \langle \psi_2 \bar{\partial} X_R \rangle$$
$$\propto \sum_{\substack{n \in H^2(X,\mathbb{Z}) \\ n = \nu \mod 2}} (n_1/R + R n_2/2) \bar{q}^{(n_1/R + R n_2/2)^2/4} q^{(n_1/R - R n_2/2)^2/4} = 0, \tag{95}$$

where, as before, $\nu \in \mathbb{Z}_2^2$ labels the discrete gauge flux. The vanishing can be attributed to anti-symmetry under the automorphism of the $H^2(X,\mathbb{Z})$ lattice acting as $n \mapsto -n$ on all the elements. For $X = \mathbb{CP}^1 \times \mathbb{CP}^1$ this automorphism is induced by the map $X \to X$ given by complex conjugation on the complex coordinates. Thus it corresponds to a certain global $\mathbb{Z}_2$ symmetry of the theory. Note that for any other spin manifold $X$ with $b_2^+ = b_2^- = 1$, the intersection form is the same as for $\mathbb{CP}^1 \times \mathbb{CP}^1$, due to the classical result about classification of even self-dual lattices. Therefore the result of the calculation will be the exactly same (namely zero) because the contribution from the boundary at infinity of the target space depends only on the intersection form and the action of the Hodge star on $H^2(X,\mathbb{R})$. However, the involution of the lattice $n \mapsto -n$ does not necessarily correspond to an orientation-preserving self-diffeomorphism of $X$.

Consider now a smooth spin 4-manifold $X$ with $b_2^+ = 1$ and $b_2^- > 1$. Note that due to Rokhlin's theorem, $b_2^- - 1 = 0 \mod 16$. If the metric on $X$ is generic, the effective 2d theory will

have $(b_2^- - 1)/4 \geq 4$ extra right-moving Majorana-Weyl fermions in addition to the fields $\psi_{1,2}$ introduced above. They originate from harmonic spinors on $X$. In this case it is not possible to saturate the corresponding fermionic zero-modes by an insertion of the supercurrent and the anomaly vanishes for a more trivial reason.

*The B Twist*

The sigma model now has $(1,1)$ supersymmetry, which for Kähler manifolds is enhanced to $(1,2)$. As before, the field content can be determined by counting harmonic forms on $X$. In the case $X = \mathbb{CP}^2$ the effective theory contains a single non-chiral non-compact boson, a single right-moving compact boson of radius $\sqrt{2}$, and their super-partners: one left-moving and two right-moving Majorana-Weyl fermions. While it is possible to saturate the zero-modes for the right-moving fermions as in the case of C twist considered above, the zero-mode for the left-moving fermion will remain unpaired, rendering the result to be zero. The same holds for other manifolds with $b_2^+ = 1$ and $b_1 = 0$.

# Acknowledgments

We would like to thank J. Manschot, G. Moore, D. Pei, B. Pioline, C. Vafa for useful discussions. EW acknowledges partial support from NSF Grant PHY-1911298.

# A    Supersymmetrization of the Four-Dimensional Action

Our goal is to determine the constants $a$ and $b$ introduced in (30). The constant $b$ will be determined in Appendix A.5 by comparison with the bosonic part of untwisted action with auxiliary fields put on-shell. The constant $a$ will be determined by identifying a bosonic term linear in the field $H$ (whose zero mode is $\mathcal{H}$) in the supersymmetrization of the bosonic Wess-Zumino term. It is convenient to work directly in the twisted theory because the superalgebra of the scalar supercharges is particularly simple.

On a four-manifold of general holonomy, the $\mathcal{N} = 4$ topological twisted theory has two scalar supercharges with an off-shell realization [1, 36, 37, 56]. For a Kähler manifold $X$, the holonomy is reduced from $SU(2)_\ell \times SU(2)_r$ to $U(1)_\ell \times SU(2)_r$. Only a $U(1)_R$ subgroup of $Spin(6)_R$ is used for twisting by replacing $U(1)_\ell$ with the diagonal $U(1)'_\ell$ of $U(1)_R \times U(1)_\ell$.

The resulting twisted theory thus has unbroken $Spin(4)_R \times U(1)_R$ global symmetry in addition to the $U(1)_{l'} \times SU(2)_r$ holonomy group. Note that since $U(1)_R$ is abelian, it remains unbroken even after turning on a non-trivial background. There are then four scalar supercharges which we denote as $Q_A$ ($A = 1, 2$) and $Q_{\dot{A}}$ ($\dot{A} = 1, 2$) and which transform as $(\mathbf{1}, \mathbf{2}, \mathbf{1})^0_{+1} \oplus (\mathbf{1}, \mathbf{1}, \mathbf{2})^0_{-1}$ respectively under $SU(2)_r \times Spin(4)_R \times U(1)'_\ell \times U(1)_R$ where the superscript denotes the $U(1)'_\ell$ charge and the subscript denotes the $U(1)_R$ charge.

## A.1    Off-shell Fields

The $\mathcal{N} = 4$ vector multiplet contains the gauge field and six scalar fields in the untwisted theory. The gauge field is not affected by twisting, and splits into the following two irreducible representations[18]:

$$A^\pm \ (\mathbf{2}, \mathbf{1}, \mathbf{1})^{\pm 1}_0 \tag{96}$$

---

[18] In the Euclidean theory, they are to be regarded as independent fields not related by complex conjugation.

corresponding to the Hodge decomposition of a 1-form on a Kähler manifold into (1,0) and (0,1) forms. Similarly, the exterior derivative splits as $d = d^+ + d^-$ :

$$d^\pm \ (\mathbf{2,1,1})_0^\pm, \tag{97}$$

where $d^+ \equiv \partial$, $d^- \equiv \bar{\partial}$ are the Dolbeault differentials. The six scalars of the untwisted theory split into three irreducible representations:

$$\Phi_{A\dot{A}} \ (\mathbf{1,2,2})_0^0,$$
$$B^{++} \ (\mathbf{1,1,1})_2^2, \qquad B^{--} \ (\mathbf{1,1,1})_{-2}^{-2}. \tag{98}$$

We have denoted the fields suppressing the $SU(2)_r$ indices to avoid clutter. The $A$ and $\dot{A}$ indices on the fields transform in the spinor and the conjugate spinor representations of $Spin(4)_R$, the superscript denotes the $U(1)'_\ell$ charge and the subscript denotes the $U(1)_R$ charge.

There are sixteen fermions which split into six irreducible representations:

$$\tilde{\psi}_A \ (\mathbf{1,2,1})_{-1}^0, \qquad \tilde{\lambda}_{\dot{A}} \ (\mathbf{1,1,2})_{+1}^0,$$
$$\tilde{\psi}_A^{--} \ (\mathbf{1,2,1})_{-1}^{-2}, \qquad \tilde{\lambda}_{\dot{A}}^{++} \ (\mathbf{1,1,2})_{+1}^{+2},$$
$$\psi_A^+ \ (\mathbf{2,2,1})_{+1}^{+1}, \qquad \lambda_{\dot{A}}^- \ (\mathbf{2,1,2})_{-1}^{-1}. \tag{99}$$

So far we have ten bosonic and sixteen fermionic fields. Taking into account gauge freedom parametrized by a single scalar bosonic field we then need seven auxiliary fields to obtain an off-shell realization of the four scalar supercharges. We introduce auxiliary fields for each fixed representation of $SU(2)_r \times U(1)'_\ell$ as follows:

$$\tilde{H}^+ \ (\mathbf{2,1,1})_{+2}^{+1}, \qquad \tilde{H}^- \ (\mathbf{2,1,1})_{-2}^{-1},$$
$$H^{++} \ (\mathbf{1,1,1})_0^{+2}, \qquad H^{--} \ (\mathbf{1,1,1})_0^{-2},$$
$$H \ (\mathbf{1,1,1})_0^0. \tag{100}$$

The fields $H^{\pm\pm}$ and $H$ can be seen to arise from a self-dual 2-form,

$$H^{(2+)} := H\omega + H^{++} + H^{--} \tag{101}$$

as the (1,1), (2,0) and (0,2) components in the Hodge decomposition on a Kähler manifold. Here and elsewhere $\omega$ denotes the Kähler form normalized such that $\int_X \omega \wedge \omega = 1$. Similarly, the fields $\tilde{H}^\pm$ combine into 1-form as its (1,0) and (0,1) components:

$$\tilde{H}^{(1)} := \tilde{H}^+ + \tilde{H}^-. \tag{102}$$

## A.2 Off-shell Superalgebra

The four supercharges are nilpotent up to gauge transformation $\delta_{\text{gauge}}(\Phi_{A\dot{B}})$ generated by $\Phi_{A\dot{B}}$:

$$\{Q_A, Q_B\} = 0, \qquad \{\bar{Q}_{\dot{A}}, \bar{Q}_{\dot{B}}\} = 0,$$
$$\{Q_A, \bar{Q}_{\dot{B}}\} = \delta_{\text{gauge}}(\Phi_{A\dot{B}}). \tag{103}$$

The off-shell realization of this algebra can be determined essentially by inspection and by comparison with the untwisted theory as we describe below.

The entire vector supermultiplet of $\mathcal{N} = 4$ theory together with the auxiliary fields splits into three multiplets of the algebra (103) satisfied by the four scalar supercharges. The transformations below are strongly constrained by the $SU(2)_r \times Spin(4)_R \times U(1)'_\ell \times U(1)_R$ global symmetry and the algebra (103). Note that the fields can always be rescaled relative to the gauge fields $A^\pm$ so that the coefficients in the supersymmetry transformations are as below.

Sections of the canonical and anti-canonical bundles belong to two short multiplets:

$$
\begin{array}{llll}
[Q_A, B^{--}] &=& \tilde{\psi}_A^{--}, & \qquad [\bar{Q}_{\dot{A}}, B^{--}] &=& 0, \\
\{Q_A, \tilde{\psi}_B^{--}\} &=& \epsilon_{AB} H^{--}, & \qquad \{\bar{Q}_{\dot{A}}, \tilde{\psi}_B^{--}\} &=& 0, \\
[Q_A, H^{--}] &=& 0, & \qquad [\bar{Q}_{\dot{A}}, H^{--}] &=& 0,
\end{array}
\tag{104}
$$

and

$$
\begin{array}{llll}
[Q_A, B^{++}] &=& 0, & \qquad [\bar{Q}_{\dot{A}}, B^{++}] &=& \tilde{\lambda}_{\dot{A}}^{++}, \\
\{Q_A, \tilde{\lambda}_{\dot{B}}^{++}\} &=& 0, & \qquad \{\bar{Q}_{\dot{A}}, \tilde{\lambda}_{\dot{B}}^{++}\} &=& \epsilon_{\dot{A}\dot{B}} H^{++}, \\
[Q_A, H^{++}] &=& 0, & \qquad [\bar{Q}_{\dot{A}}, H^{++}] &=& 0,
\end{array}
\tag{105}
$$

The remaining fields including the gauge field form a long multiplet:

$$
\begin{array}{llll}
\{Q_A, \lambda_{\dot{B}}^-\} &=& d^- \Phi_{A\dot{B}}, & \qquad \{\bar{Q}_{\dot{A}}, \psi_B^+\} &=& d^+ \Phi_{B\dot{A}}, \\
[Q_A, \Phi_{B\dot{B}}] &=& \epsilon_{AB} \tilde{\lambda}_{\dot{B}}, & \qquad [\bar{Q}_{\dot{A}}, \Phi_{B\dot{B}}] &=& -\epsilon_{\dot{A}\dot{B}} \tilde{\psi}_B, \\
\{Q_A, \tilde{\lambda}^{\dot{B}}\} &=& 0, & \qquad \{\bar{Q}_{\dot{A}}, \tilde{\psi}^B\} &=& 0,
\end{array}
$$

$$
\begin{array}{llll}
[Q_A, \tilde{H}^-] &=& -d^- \tilde{\psi}_A, & \qquad [\bar{Q}_{\dot{A}}, \tilde{H}^+] &=& d^+ \tilde{\lambda}_{\dot{A}}, \\
\{Q_A, \tilde{\psi}_B\} &=& \epsilon_{AB} H, & \qquad \{\bar{Q}_{\dot{A}}, \tilde{\lambda}_{\dot{B}}\} &=& \epsilon_{\dot{A}\dot{B}} H, \\
[Q_A, H] &=& 0, & \qquad [\bar{Q}_{\dot{A}}, H] &=& 0,
\end{array}
\tag{106}
$$

$$
\begin{array}{llll}
[Q_A, A^+] &=& \psi_A^+, & \qquad [\bar{Q}_{\dot{A}}, A^-] &=& \lambda_{\dot{A}}^-, \\
\{Q_A, \psi_B^+\} &=& \epsilon_{AB} \tilde{H}^+, & \qquad \{\bar{Q}_{\dot{A}}, \lambda_{\dot{B}}^-\} &=& \epsilon_{\dot{A}\dot{B}} \tilde{H}^-, \\
[Q_A, \tilde{H}^+] &=& 0, & \qquad [\bar{Q}_{\dot{A}}, \tilde{H}^-] &=& 0,
\end{array}
$$

$$
\begin{array}{llll}
[Q_A, A^-] &=& 0, & \qquad [\bar{Q}_{\dot{A}}, A^+] &=& 0,
\end{array}
$$

where we have grouped the fields into shorter multiplets with respect to the $Q_A$ or $\bar{Q}_{\dot{A}}$ supercharges separately. The fields $\Phi, \tilde{\psi}, \tilde{\lambda}, H$ form a submultiplet of the superalgebra (103).

The normalization of the gauge field can be fixed by specifying the coefficient for the kinetic term. The realization of the superalgebra above still leaves the freedom to rescale other fields together with supercharges without changing the supersymmetry transformations:

$$
\begin{array}{c}
A^\pm \to A^\pm, \ Q \to CQ, \ \bar{Q} \to C\bar{Q}, \ \psi^+ \to C\psi^+, \ \lambda^- \to C\psi^-, \\
\tilde{H}^\pm \to C^2 \tilde{H}^\pm, \ \Phi \to C^2 \Phi, \ \tilde{\psi} \to C^3 \tilde{\psi}, \ \tilde{\lambda} \to C^3 \tilde{\lambda}, \ H \to C^4 H, \\
B^{++} \to C^2 B^{++}, \ \tilde{\lambda}^{++} \to C^3 \tilde{\lambda}^{++}, \ H^{++} \to C^4 H^{++}, \\
B^{--} \to C^2 B^{--}, \ \tilde{\psi}^{--} \to C^3 \tilde{\psi}^{--}, \ H^{--} \to C^4 H^{--}.
\end{array}
\tag{107}
$$

### A.3 Supersymmetrization of the Free Action

The bosonic part of the free action can be fixed by requiring that one obtains the standard kinetic term for the gauge field for the unbroken $U(1)$ subgroup of $SO(3)$ on the Coulomb branch after eliminating the auxiliary fields:

$$
S_{4d}^{\text{free}} = 2\pi i \tau \int_X \frac{F_A \wedge F_A}{4\pi^2} - \frac{\tau_2}{\pi} \int_X H^{(2+)} \wedge (H^{(2+)} - 2F_A) + \dots
\tag{108}
$$

Using Hodge decomposition (101) of $H^{(2+)}$, the remaining terms are fixed by supersymmetry:

$$
\begin{aligned}
S_{4d}^{\text{free}} = {}& 2\pi i \tau \int_X \frac{F_A \wedge F_A}{4\pi^2} + \frac{2\tau_2}{\pi} \int_X \Big( -\frac{\omega \wedge \omega}{2} H^2 + d^+ \psi_A^+ \tilde{\psi}^{--A} + d^- \lambda_{\dot{A}}^- \tilde{\lambda}^{++\dot{A}} + \\
& \omega \wedge [H(d^+ A^- + d^- A^+) + \frac{1}{2} d^+ \Phi^{A\dot{A}} d^- \Phi_{A\dot{A}} + d^+ \tilde{\lambda}^{\dot{A}} \lambda_{\dot{A}}^- + \psi_A^+ d^- \tilde{\psi}^A + \tilde{H}^+ \tilde{H}^-] \\
& \qquad -H^{++} H^{--} + d^+ A^+ H^{--} + d^- A^- H^{++} + \tilde{H}^+ d^+ B^{--} + \tilde{H}^- d^- B^{++} \Big).
\end{aligned}
\tag{109}
$$

One can show that the term in (109) proportional to $\tau_2$ is $Q$-exact. Using $Spin(4)_R$ R-symmetry, one can choose $Q$ to be a linear combination of the form

$$Q = \alpha Q_1 + \beta \bar{Q}_1, \tag{110}$$

where $\alpha$ and $\beta$ are constants. Define

$$\Lambda^1 := \frac{i}{\pi} \int_X \left( \frac{\omega \wedge \omega}{2} H\tilde{\psi}_2 + \omega \wedge [-(d^+A^- + d^-A^+)\tilde{\psi}_2 + \tilde{H}^-\psi_2^+ + d^+\Phi^{1\dot{B}}\lambda_{\dot{B}}^-] \right.$$
$$\left. -2d^+A^+\tilde{\psi}_2^{--} + H^{++}\tilde{\psi}_2^{--} + 2d^+B^{--}\psi_2^+ \right) \tag{111}$$

and

$$\bar{\Lambda}^1 := \frac{i}{\pi} \int_X \left( \frac{\omega \wedge \omega}{2} H\tilde{\lambda}_2 + \omega \wedge [-(d^+A^- + d^-A^+)\tilde{\lambda}_2 - \tilde{H}^+\lambda_2^- - d^-\Phi^{B1}\psi_B^+] \right.$$
$$\left. -2d^-A^-\tilde{\lambda}_2^{++} + H^{--}\tilde{\lambda}_2^{++} + 2d^-B^{++}\lambda_2^- \right). \tag{112}$$

They satisfy

$$\{\bar{Q}_1, \Lambda^1\} = 0, \qquad \{Q_1, \bar{\Lambda}^1\} = 0, \tag{113}$$

and[19]

$$S_{4d}^{free} = 2\pi i \tau \int_X \frac{F_A \wedge F_A}{4\pi^2} - i\tau_2(\{Q_1, \Lambda^1\} + \{\bar{Q}_1, \bar{\Lambda}^1\}). \tag{114}$$

It follows that

$$\frac{\partial S_{4d}^{free}}{\partial \bar{\tau}} = \{Q, \Lambda\}, \tag{115}$$

where

$$\Lambda = \frac{1}{2\alpha}\Lambda^1 + \frac{1}{2\beta}\bar{\Lambda}^1. \tag{116}$$

For convenience one can choose $\alpha = \beta = 1$ as in Section 3.3.

## A.4 Supersymmetrization of the Wess-Zumino Term

To determine the coefficient $a$ of the term linear in $\mathcal{H}$ in the zero mode action (30), it is necessary to determine the terms in the full action that are linear in the field $H$. In this subsection we show that such a term is indeed present and is necessary to cancel supersymmetric variation of the bosonic Wess Zumino term (21).

We start by writing the Wess Zumino term (21) in the twisted field variables. The topological twist on a Kähler $X$ is realized by turning on background field strength for the subgroup of R-symmetry $Spin(2)_R \subset Spin(6)_R$. One can assume that the $Spin(2)_R$ rotates in the 4-5 plane in the $\mathbb{R}^6$ field space of scalars in the untwisted theory. With this choice, one can relate the fields $\Phi^I, I = 0 \ldots 5$ with the bosonic fields in (97) as follows:

$$\begin{aligned}
\Phi^{11} &= \Phi^2 + i\Phi^3, & \Phi^{22} &= \Phi^2 - i\Phi^3, \\
\Phi^{12} &= \Phi^0 + i\Phi^1, & \Phi^{21} &= -\Phi^0 + i\Phi^1, \\
B^{++} &= \Phi^4 + i\Phi^5, & B^{--} &= \Phi^4 - i\Phi^5.
\end{aligned} \tag{117}$$

Only nonzero components of $F^{I_1 I_2}$ in (21) are $F^{45} = -F^{54}$, related to the curvature of the canonical bundle when restricted to $X = \mathbb{CP}^2$. Let $\omega$ be the Kähler form on $\mathbb{CP}^2$ normalized such that $\int_{\mathbb{CP}^1} \omega = 1$ for a standard embedding of $\mathbb{CP}^1$ into $\mathbb{CP}^2$. Since the canonical bundle

---

[19]Note that for the action restricted to the supermultiplet (106) it is possible to use just $Q_1$ or $\bar{Q}_1$.

is $O(-3)$ bundle one has $c_1(K\mathbb{CP}^2) = -3[\omega]$. Moreover, $F^{45}|_X = 3 \cdot 2\pi \, \omega$ when the connection on $K\mathbb{CP}^2$ is induced by Levi-Civita connection on $T\mathbb{CP}^2$ for the Fubini-Study metric.

Since $\Phi^{A\dot{A}}$ and $\Phi^4 \pm i\Phi^5$ belong to different supermultiplets (104) and (106) after twisting, one can restrict $\Phi^4$ and $\Phi^5$ to be zero. With the field redefinitions above the bosonic Wess-Zumino term $S_{\text{4d WZ}}$ equals

$$\frac{i}{\pi} \int_{\Xi^5} \frac{F^{45}}{2\pi} \wedge \frac{\Phi^{11}d\Phi^{22}d\Phi^{12}d\Phi^{21} - \Phi^{22}d\Phi^{12}d\Phi^{21}d\Phi^{11} + \Phi^{12}d\Phi^{21}d\Phi^{11}d\Phi^{22} - \Phi^{21}d\Phi^{11}d\Phi^{22}d\Phi^{12}}{|\Phi|^4}$$

(118)

where[20] $|\Phi|^2 := \Phi_{A\dot{A}}\Phi^{A\dot{A}} = 2(\Phi^{11}\Phi^{22} - \Phi^{12}\Phi^{21}) = 2((\Phi^0)^2 + (\Phi^1)^2 + (\Phi^2)^2 + (\Phi^3)^2)$.

It turns out that (118) by itself cannot be supersymmetrized. One can consider instead

$$S'_{\text{WZ}} = S_{\text{4d WZ}} + S_{\text{grav}},$$

(119)

with

$$S_{\text{grav}} = -\frac{3i}{2\pi} \int_X \omega \wedge \frac{d^-\Phi_{A\dot{A}}d^+\Phi^{A\dot{A}}}{|\Phi|^2},$$

(120)

which *can* be supersymmetrized to the desired order. This term can arise from a term in the effective action in the untwisted theory of the form

$$S_{\text{grav}} = \int d^4x \sqrt{g} \, (c_1 R g^{\mu\nu} + c_2 R^{\mu\nu}) \frac{1}{\|\Phi\|^2} \sum_{I=1}^{6} \partial_\mu \Phi^I \partial_\nu \Phi^I$$

(121)

for some constants $c_1$ and $c_2$, and can be thought of as a non-minimal coupling to gravity required by supersymmetry on a curved manifold. It respects both scaling and $Spin(6)_R$ symmetry and reduces to (120) on $\mathbb{CP}^2$ when $\Phi^4 = \Phi^5 = 0$. Such a term is indeed known to be present as a four derivative coupling in $\mathcal{N} = 2$ supergravity [57].

Our goal is to determine whether a bosonic term linear in $H$ is required by supersymmetry. Since we are trying to relate a bosonic term to another bosonic term, we will have to consider at least two consecutive supersymmetry transformations. A general variation $S'_{\text{WZ}}$ is given by

$$\delta S'_{\text{WZ}} = \int_X \frac{3i\omega}{\pi} \wedge \frac{(2\Phi^{A\dot{A}}d^-\Phi_{B\dot{A}}d^+\Phi_{A\dot{B}} - |\Phi|^2 d^+ d^-\Phi_{B\dot{B}})\delta\Phi^{B\dot{B}}}{|\Phi|^4}.$$

(122)

Even though the Wess-Zumino action is defined by a 5d integral, its variation must be a local 4d integral. The supersymmetry variation can be canceled by adding the term

$$S_1 = \int_X \frac{3i\omega}{\pi} \wedge \left[ \frac{2\Phi^{A\dot{A}}d^+\Phi_{A\dot{B}}\lambda^-_{\dot{A}}\tilde{\lambda}^{\dot{B}}}{|\Phi|^4} + \frac{d^+\lambda^-_{\dot{A}}\tilde{\lambda}^{\dot{A}}}{|\Phi|^2} + \frac{2\Phi^{A\dot{A}}d^-\Phi_{B\dot{A}}\psi^+_A\tilde{\psi}^B}{|\Phi|^4} + \frac{d^-\psi^+_A\tilde{\psi}^A}{|\Phi|^2} \right].$$

(123)

With some algebra, one can verify that the variation of $S'_{\text{WZ}}$ with respect to $Q_C$ is completely canceled by the variation of the first two terms with respect to $Q_C$; and the variation with respect to $\bar{Q}_{\dot{C}}$ is completely canceled by the variation of the last two terms with respect to $\bar{Q}_{\dot{C}}$.

However the variation of the first two terms (123) with respect to $\bar{Q}_{\dot{C}}$ and the last two terms with respect to $Q_C$ is nonzero. Restricting to one fermion terms modulo terms linear in $\tilde{H}^-$, one can verify that this variation can be canceled by the variation of the term

$$S_2 = \int_X \frac{3i\omega}{\pi} \wedge \left[ \frac{2\Phi^{A\dot{A}}\psi^+_A\lambda^-_{\dot{A}}H}{|\Phi|^4} + \frac{F_A H}{|\Phi|^2} \right].$$

(124)

---

[20]Note that $|\Phi|^2 = 2\|\Phi\|^2|_{\Phi^4 = \Phi^5 = 0}$.

We have thus concluded that a term linear in $F_A$ and linear in $H$ is necessarily present in the supersymmetrization of the Wess-Zumino action. Therefore, the coefficient of this term and in turn $a$ is topological in origin and is determined by the anomaly matching which fixes the coefficient of the Wess-Zumino term.

## A.5 Restriction to Zero-Modes

Given the relevant terms in the four-dimensional action, one can determine the effective action over zero-modes to the desired order by restricting the four-dimensional fields to the zero-modes. When $X = \mathbb{CP}^2$, the canonical bundle has no harmonic sections and $b_1 = 0$. The only zero-modes arise from fields in representations of the form $(\mathbf{1}, *, *)^{\mathbf{0},*}$ and hence are given by:

$$
\begin{aligned}
\Phi^{A\dot{A}}|_{\text{zm}} &= u^{A\dot{A}}, & \tilde{\psi}^A|_{\text{zm}} &= \chi^A, \\
\tilde{\lambda}^{\dot{A}}|_{\text{zm}} &= \bar{\chi}^{\dot{A}}, & H|_{\text{zm}} &= \mathcal{H}.
\end{aligned}
\tag{125}
$$

The supercharges in (106), restricted to the zero-modes, then can be realized as the following linear differential operators:

$$
\begin{aligned}
Q_A|_{\text{zm}} &= \mathcal{H}\frac{\partial}{\partial \chi^A} + \bar{\chi}^{\dot{A}}\frac{\partial}{\partial u^{A\dot{A}}}, \\
\bar{Q}_{\dot{A}}|_{\text{zm}} &= \mathcal{H}\frac{\partial}{\partial \bar{\chi}^{\dot{A}}} - \chi^A\frac{\partial}{\partial u^{A\dot{A}}}.
\end{aligned}
\tag{126}
$$

The restriction of (109) to the zero-modes is then given by

$$
S_{\text{4d}}^{\text{free}}|_{\text{zm}} = -\frac{\tau_2}{\pi}\mathcal{H}(\mathcal{H} - 4\pi n) + 2\pi i \tau n^2,
\tag{127}
$$

where we have used the fact that

$$
\int_{\mathbb{CP}^1} \frac{F_A}{2\pi} = n \in \frac{1}{2}\mathbb{Z}.
\tag{128}
$$

The restriction of (116) is then given by

$$
\Lambda|_{\text{zm}} = -\frac{i}{4\pi}(\chi^1 + \bar{\chi}^1)(\mathcal{H} - 4\pi n).
\tag{129}
$$

The reduction of the term (124) in the effective action to the zero-modes reads

$$
S_2|_{\text{zm}} = \frac{6i\,n\mathcal{H}}{|u|^2}.
\tag{130}
$$

It follows that the constant $a$ defined in (30) is equals $3/\pi$.

The coefficient $b$ in (30) of the term quadratic in $H$ cannot be fixed by the supersymmetrization of the Wess-Zumino term by the four supercharges considered above. This can be seen by considering a term in the action of the following form motivated by (26):

$$
\int_X \omega \wedge \omega \left( H^2 f(|\Phi|^2, H) + 2\Phi_{A\dot{A}} H f'(|\Phi|^2, H)\tilde{\psi}^A\tilde{\lambda}^{\dot{A}} \right.
$$
$$
\left. + (2f'(|\Phi|^2, H) + |\Phi|^2 f''(|\Phi|^2, H))\epsilon_{AB}\epsilon_{\dot{A}\dot{B}}\tilde{\psi}^A\tilde{\psi}^B\tilde{\lambda}^{\dot{A}}\tilde{\lambda}^{\dot{B}} \right), \quad (131)
$$

where $f$ is any function and prime denotes the derivative with respect to the first argument. This term is annihilated by all four supercharges and hence is not fixed by the analysis in this appendix. When restricted to zero-modes, such a term will shift the value of $b$.

To determine $b$, one can appeal to the untwisted on-shell action. If $b \neq -1$, then after integrating the auxiliary field $H$, there must be a term proportional to

$$\int_X \frac{F_A^+ F_A^+ \, \omega}{|\Phi|^2} \tag{132}$$

on $X = \mathbb{CP}^2$ with a nonzero coefficient. The indices in $F_A^+ F_A^+ \, \omega$ are contracted appropriately to obtain a 4-form. Such a term can arise from only two possible terms.

- A term of the form

$$\int \frac{F_A^+ F_A^+ F_R}{\|\Phi\|^2} \tag{133}$$

  in the theory in flat space, where $F_R$ is the background R-symmetry. However, such a term can be invariant only under $Spin(4)_R \times U(1)_R$ but cannot be made invariant under the full $Spin(6)_R$ R-symmetry of the untwisted theory.

- A term of the form

$$\int \frac{F_A^+ F_A^+ R}{\|\Phi\|^2} \tag{134}$$

  where $R$ is the Riemann curvature tensor. The scalar fields $\Phi$ in the term above can be restricted to be purely in the $\mathcal{N} = 2$ vector multiplet. However, it is known that there is no such 4-derivative coupling in $\mathcal{N} = 2$ supergravity ( [57]).

Since neither term is consistent with superymmetry and R-symmetry, we conclude $b = -1$.

# B $\quad SO(3)$ versus $U(2)$

In this section we comment on the relation between the partition functions for the $SO(3)$ and $U(2)$ gauge groups. From the point of view of 6d (2,0) theory, the $\mathfrak{u}(2)$ Lie algebra is in a certain sense more natural because it describes the stack of two M5-branes without removing the centre of mass degrees of freedom. Moreover, the six-dimensional $\mathfrak{u}(2)$ (2,0) theory is *absolute* because it has a self-dual lattice of string charges and thus has a single partition function. By contrast, the $\mathfrak{su}(2)$ (2,0) theory is relative and has a vector of partition functions labeled by discrete fluxes. On $X \times T^2$ partition function should transform to itself under modular transformations, up to a phase related to 't Hooft anomalies.

Let us consider how the effective two-dimensional theory in the $\mathfrak{u}(2)$ case is modified compared to $\mathfrak{su}(2)$ case. The analysis in Section 4.2 can be repeated for the 6d tensor multiples valued in the Cartan sub algebra of $\mathfrak{u}(2)$. In particular, the KK reduction of the self-dual 2-form field $B$ now leads to $\widehat{\mathfrak{u}(2)}_1$ right-moving WZW CFT, instead of $\widehat{\mathfrak{su}(2)}_1 \cong \widehat{\mathfrak{u}(1)}_2$. This two-dimensional theory is now also absolute[21] and its character is

$$\bar{\chi}_{0,0}^{\widehat{\mathfrak{u}(2)}_1}(\tau; z) = \frac{1}{\eta(\tau)^2} \sum_{n \in \mathbb{Z}^2} \bar{q}^{\frac{n_1^2 + n_2^2}{2} + (n_1 + n_2)\bar{z}} = \left[ \overline{\frac{\vartheta_3(\tau; z)}{\eta(\tau)}} \right]^2, \tag{135}$$

which again captures contribution of abelian instantons. We have included the fugacity $e^z$ for the diagonal $\mathfrak{u}(1)$. From the 4d perspective such refinement is realized by adding the topological term $z \int_{\mathbb{CP}^1} c_1$ to the action on $\mathbb{CP}^2$. The decoupling of the diagonal $\mathfrak{u}(1)$, corresponding to the center of mass degrees of freedom, is then done via the following decomposition of characters:

$$\bar{\chi}_{0,0}^{\widehat{\mathfrak{u}(2)}_1}(\tau; z) = \bar{\chi}_0^{\widehat{\mathfrak{u}(1)}_2}(\tau; z) \bar{\chi}_0^{\widehat{\mathfrak{su}(2)}_1}(\tau) + \bar{\chi}_1^{\widehat{\mathfrak{u}(1)}_2}(\tau; z) \bar{\chi}_1^{\widehat{\mathfrak{su}(2)}_1}(\tau), , \tag{136}$$

---

[21]As a spin-theory, as we will comment on below.

where

$$\bar{\chi}_\lambda^{\widehat{u(1)}_2}(\tau;z) := \frac{1}{\eta(\tau)} \sum_{n\in\mathbb{Z}} \bar{q}^{(n+\lambda/2)^2+(2n+\lambda)\bar{z}}. \tag{137}$$

The characters (47) transform as a dimension 2 complex representation of $SL(2,\mathbb{Z})$, up to an overall multiplier system related to the gravitational anomaly. The $S$ and $T$ elements are represented by the following matrices:

$$S = \begin{pmatrix} 1 & 1 \\ 1 & -1 \end{pmatrix}, \qquad T = e^{\frac{\pi i}{12}} \begin{pmatrix} 1 & 0 \\ 0 & i \end{pmatrix}. \tag{138}$$

The left hand side of (136) is a modular function (again, with a multiplier system) under the index 3 subgroup $\langle S, T^2 \rangle \subset SL(2,\mathbb{Z})$ generated by $S$ and $T^2$. This is expected because $U(2)$ is self-dual, and on a non-spin manifold the $U(2)$ theory is only invariant under the shift of the theta-angle by $4\pi$. From the 2d point of view $\langle S, T^2 \rangle$ is the subgroup of $SL(2,\mathbb{Z})$ preserving the antiperiodic-antiperiodic spin structure on $T^2$. This is the spin structure for which the character (135) is the partition function of $\widehat{u(2)}_1$ chiral WZW, which should be considered as a spin theory. Note that the matrices $S$ and $T^2$ transforming the characters of $\widehat{\mathfrak{su}(2)}_1$, or equivalently $\widehat{u(1)}_2$, happen to be real (up to an overall phase), so the representation is self-conjugate. In particular this means that formally one could also take instead a linear combination

$$\chi_0^{\widehat{u(1)}_2}(\tau;z)\bar{\chi}_0^{\widehat{\mathfrak{su}(2)}_1}(\tau) + \chi_1^{\widehat{u(1)}_2}(\tau;z)\bar{\chi}_1^{\widehat{\mathfrak{su}(2)}_1}(\tau) \tag{139}$$

to achieve the same effect, that is to produce an $\langle S, T^2 \rangle$ Jacobi modular function (with a different multiplier system). Therefore there is no contradiction with the discussion in Section 3.1.

As an aside we note that if one considers periodic-periodic spin structure instead, the partition function of the chiral $\widehat{u(2)}_1$ WZW reads

$$\bar{\chi}_{1,1}^{\widehat{u(2)}_1}(\tau;z) = \frac{1}{\eta(\tau)^2} \sum_{n\in\mathbb{Z}^2} (-1)^{n_1+n_2+1} \bar{q}^{\frac{(n_1+1/2)^2+(n_2+1/2)^2}{2}+(n_1+n_2+1)\bar{z}} = \left(\overline{\frac{\vartheta_1(z;\tau)}{\eta(\tau)}}\right)^2. \tag{140}$$

It is modular function of the full $SL(2,\mathbb{Z})$ group, since it preserves the unique odd spin structure. From the 4d point of view the diagonal of $U(2)$ gauge group is replaced by a Spin$^c$ structure, which results in half-integer shifts of the fluxes on the non-spin manifold $\mathbb{CP}^2$. This corresponds to Freed-Witten anomaly in string/M-theory setting. The character decomposition in this case reads

$$\bar{\chi}_{1,1}^{\widehat{u(2)}_1}(\tau;z) = \bar{\chi}_0^{\widehat{u(1)}_2}(\tau;z)\bar{\chi}_1^{\widehat{\mathfrak{su}(2)}_1}(\tau) - \bar{\chi}_1^{\widehat{u(1)}_2}(\tau;z)\bar{\chi}_0^{\widehat{\mathfrak{su}(2)}_1}(\tau). \tag{141}$$

## C Closed Forms

### C.1 Closed 3-forms on $\mathbb{R}^4 \setminus \{0\}$

Let $v^i, i = 0, \ldots, 3$ be the standard coordinates on $\mathbb{R}^4$. It is easy to see that the 3-form

$$\omega_3 := \frac{v^0 dv^1 dv^2 dv^3 - v^1 dv^2 dv^3 dv^0 + v^2 dv^3 dv^0 dv^1 - v^3 dv^0 dv^1 dv^2}{2\pi^2 \|v\|^2} \in \Omega^3(\mathbb{R}^4 \setminus \{0\}) \tag{142}$$

is closed, $SO(4)$ invariant and normalized such that

$$\int_{\Sigma^3} \omega_3 = 1 \tag{143}$$

for any 3-cycle $\Sigma_3$ which represents the generator of $H_3(\mathbb{R}^4 \setminus \{0\}, \mathbb{Z}) \cong \mathbb{Z}$, in particular for a round 3-sphere centered at the origin. Thus, $\omega_3$, can be understood as the pullback of the standard volume form (with unit volume) on $S^3$ with respect to the homotopy equivalence

$$
\begin{aligned}
\mathbb{R}^4 \setminus \{0\} &\longrightarrow S^3, \\
v &\longmapsto v/\|v\|.
\end{aligned}
\tag{144}
$$

Under the change of variables

$$
\begin{aligned}
u^{11} &= v^2 + iv^3 & u^{22} &= v^2 - iv^3 \\
u^{12} &= v^0 + iv^1 & u^{21} &= -v^0 + iv^1
\end{aligned}
\tag{145}
$$

the 3-form above becomes

$$
\omega_3 = \frac{u^{11}du^{22}du^{12}du^{21} - u^{22}du^{12}du^{21}du^{11} + u^{12}du^{21}du^{11}du^{22} - u^{21}du^{11}du^{22}du^{12}}{2\pi^2 |u|^2},
\tag{146}
$$

where $|u|^2 := \epsilon_{AB}\epsilon_{\dot{A}\dot{B}}u^{A\dot{A}}u^{B\dot{B}} = 2(u^{11}u^{22} - u^{12}u^{21})$. Let

$$
\zeta_3 := \frac{(u^{12}du^{11} - u^{11}du^{12})du^{22}du^{21}}{|u|^4} \in \Omega^3(\mathbb{R}^4 \setminus \{0\}),
\tag{147}
$$

and

$$
\tilde{\zeta}_3 := \frac{(u^{21}du^{11} - u^{11}du^{21})du^{12}du^{22}}{|u|^4} \in \Omega^3(\mathbb{R}^4 \setminus \{0\}).
\tag{148}
$$

It is easy to see that these 3-forms are also closed and, moreover,

$$
\begin{aligned}
\zeta_3 &= \pi^2 \omega_3 + d(\ldots), \\
\tilde{\zeta}_3 &= \pi^2 \omega_3 + d(\ldots),
\end{aligned}
\tag{149}
$$

that is $\int_{\Sigma_3} \zeta_3 = \int_{\Sigma_3} \tilde{\zeta}_3 = \pi^2$ any 3-cycle $\Sigma_3$ which generates $H_3(\mathbb{R}^4 \setminus \{0\}, \mathbb{Z}) \cong \mathbb{Z}$.

## C.2 The Euler Angular Form $\eta_4$

We now verify explicitly that the Euler angular form $\eta_4$ is closed when the background R-symmetry connection is turned on only for a subgroup $Spin(2) \subset Spin(5)$ relevant for the topological twisting on Kähler 4-manifolds. One can assume that $Spin(2)$ corresponds to rotations in the 4-5 plane and substitute $A^{45} = -A^{54}$ and $A^{ab} = 0$ otherwise in (42) to obtain

$$
\eta_4 = \frac{4!}{64\pi^2} \Big\{ Z^1 dZ^2 dZ^3 DZ^4 DZ^5 + Z^2 dZ^3 DZ^4 DZ^5 dZ^1 + \ldots + Z^5 dZ^1 dZ^2 dZ^3 DZ^4
$$
$$
- \frac{1}{3}dA(Z^1 dZ^2 dZ^3 + Z^2 dZ^3 dZ^1 + Z^3 dZ^2 dZ^1) \Big\}.
\tag{150}
$$

Let

$$
\omega_4 := \frac{1}{64\pi^2} \epsilon_{a_1 a_2 a_3 a_4 a_5} Z^{a_1} dZ^{a_2} dZ^{a_3} dZ^{a_4} dZ^{a_5}
\tag{151}
$$

be the pullback of the standard volume form on $S^4$ normalized such that $\int_{S^4} \omega_4 = 1$. Using $DZ^4 DZ^5 = dZ^4 dZ^5 - A(Z^4 dZ^4 + Z^5 dZ^5)$ and $\sum_{i=1}^5 (Z^i)^2 = 1$ we have

$$
\begin{aligned}
\eta_4 = {} & \omega_4 + \frac{4!}{64\pi^2} \Big\{ -A(Z^4 dZ^4 + Z^5 dZ^5)(Z^1 dZ^2 dZ^3 + Z^2 dZ^3 dZ^1 + Z^3 dZ^1 dZ^2) \\
& \qquad\qquad + A((Z^4)^2 + (Z^5)^2) dZ^1 dZ^2 dZ^3 \\
& \qquad\qquad - \frac{1}{3} dA(Z^1 dZ^2 dZ^3 Z^2 dZ^3 dZ^1 + Z^3 dZ^2 dZ^1) \Big\} \\
= {} & \omega_4 + \frac{4!}{64\pi^2} \Big\{ A(Z^1 dZ^1 + Z^2 dZ^2 + Z^3 dZ^3)(Z^1 dZ^2 dZ^3 + Z^2 dZ^3 dZ^1 + Z^3 dZ^1 dZ^2) \\
& \qquad\qquad + A(1 - (Z^1)^2 - (Z^2)^2 - (Z^3)^2) dZ^1 dZ^2 dZ^3 \\
& \qquad\qquad - \frac{1}{3} dA(Z^1 dZ^2 dZ^3 + Z^2 dZ^3 dZ^1 + Z^3 dZ^2 dZ^1) \Big\} \\
= {} & \omega_4 + \frac{4!}{64\pi^2} \Big\{ A dZ^1 dZ^2 dZ^3 - \frac{1}{3} dA(Z^1 dZ^2 dZ^3 + Z^2 dZ^3 dZ^1 + Z^3 dZ^2 dZ^1) \Big\} \\
= {} & \omega_4 - \frac{1}{8\pi^2} d \Big\{ A \big( Z^1 dZ^2 dZ^3 + Z^2 dZ^3 dZ^1 + Z^3 dZ^2 dZ^1 \big) \Big\} \quad (152)
\end{aligned}
$$

from which it follows that $d\eta_4 = 0$.

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
