# Peer review of "Duality and Mock Modularity"

_SciPost Physics, doi:SciPost Phys. 9, 072 (2020)_

## Round 2 · Referee Report · Anonymous (Referee 1) · 2020-9-5

Report

The paper is surely top 10% and to be published.

The topic is very interesting and opens a new and simple approach to the interpretation of the modular properties of Vafa-Witten partition functions in terms of a new holomorphic anomaly equation for the four dimensional gauge theory.
The solution of the problem in terms of anti-instantons contributions is elegant and very natural from the physics view point.

In the text the contribution of point-like instantons, which is known to the experts, is not explained as carefully as it could to make the text accessible at a broader level.

---

## Round 2 · Referee Report · Anonymous (Referee 2) · 2020-10-9

Report

The partition function of twisted $N=4$ super Yang-Mills (SYM) theory on a manifold $X$ was studied in a paper by Vafa and Witten in 1994. It was calculated for a number of examples of $X$ in that paper by reducing the path integral to an integral over the moduli space of instantons summed over all instanton numbers. For all $X$ the partition function $Z(\tau)$ calculated in this manner is a holomorphic function.

S-duality of $N=4$ SYM implies that the Vafa-Witten partition function should have good modular properties. In particular, for gauge group $SO(3)$ the partition function for $X=\mathbb{C} \mathbb{P}^2$ is expected to be invariant under the action of a certain subgroup of $SL(2,\mathbb{Z})$ on the gauge coupling $\tau$. On the other hand, the above-mentioned calculation in this case gave rise to a function $Z(\tau)$ which is not a modular function, and therefore seems to be in conflict with S-duality of the theory. It was recognized in the Vafa-Witten paper that the function has (what is today called) a mock modular property -- it can be completed to a modular function by adding a specific non-holomorphic correction term to $Z$, thus giving rise to a holomorphic anomaly. However, the physical origin of this correction term remained mysterious.

In the current paper the authors resolve this puzzle by explaining the physical origin of mock modularity. Using this idea they calculate the holomorphic anomaly -- in two different ways. The first way is a careful reconsideration of the Coulomb branch integral (à la Moore-Witten) for this theory; the boundary terms in this integral precisely reproduce the holomorphic anomaly. The second way is by constructing a dual two-dimensional sigma model with non-compact target space which also has the same holomorphic anomaly. The 2d and the 4d theories are related in that they both arise from the same 6d $(2,0)$ theory on an M5 brane in different limits.

The paper contains new interesting results, and it is very clearly written. It will pay dividends to those who read it with some care -- along the way there is also a nice review of the three possible twistings of $N=4$ SYM, of the brane construction of the theory, and an explicit construction of the off-shell supersymmetry algebra that is needed for the localization calculations. A final section contains some generalizations of this story to other $X$ and other gauge groups. I have no doubt that it should be published.

I only have a few minor remarks:

-- At various places in the introduction the phrase "modular function" (or "mock modular function") is used. I suppose the authors mean "a function which has modular properties".
This may give rise to a small confusion since the phrase also has a definition (invariant under modular transformation) which does not hold in the general context as used in the paper (e.g. for K3 the answer is a power of $\eta(\tau)$). I would suggest using (mock) modular form---indeed, later in the paper this is used by the authors---or inserting a note saying that they are using the phrase loosely.

-- Is it clear that there is no boundary term at $u=0$ in Section 3.3? (Footnote 9 seems to say that they need to cut out the origin, is this regulator consistent with the symmetries?) I would imagine so, but a brief note of explanation would be helpful.

-- Typo: it should be $(2,0)$ in the penultimate paragraph on Page 11.

-- It may be worth reminding the reader that $KX$ at the beginning of Section 5 stands for the canonical bundle of X (as defined earlier in the paper).

-- One thing that confused me a bit is the penultimate paragraph on Page 5. The authors say that (a) there is full control of the 2d theory including the overall normalization (using a relation to the Hamiltonian interpretation), (b) the overall normalization is not under control in the 4d theory, and (c) the two theories come from the same 6d theory. If we track the 2d/4d duality through the M5 brane very carefully, can we keep track of the normalization, or at what step do we lose track of the normalization?
(This last point is probably just something I didn't understand personally, and this should not hold up publication of the paper but I would be happy if the authors could answer this.)

---

## Round 3 · Referee Report · Anonymous · 2020-10-23

Report

The authors have addressed all my previous comments satisfactorily.
In particular, in the normalization as defined in Footnote 2 the partition function for K3 has weight 0 and not -12 (as the partition function Z in the Vafa-Witten paper), so there is no confusion about the notation now.

---

## Round 3 · Referee Report · Anonymous · 2020-10-26

Report

The second version includes a change in the direction I suggested.

---

## Round 3 · Author Response

We thank the referees for the comments. We have addressed them as indicated in the ``list of changes''.

---

## Round 3 · List of Changes

In reply to Report 2:

1) A clarifying Footnote 2 added on page 2.

2) The text in the former Footnote 10 (which contained some relevant comments) was expanded and moved to the main text below eq. (3.29) on page 21.

3) The typo is corrected, we thank the referee for catching it.

4) A reminder about notation $KX$ is added on page 33.

5) Clarifying comments added on page 21.

In reply to Report 1:

6) A paragraph with a brief review of point-like instantons added on page 21.

---

## Editorial Decision

published